# Targeting Pan-Cancer Stemness: Core Regulatory lncRNAs as Novel Therapeutic Vulnerabilities

**DOI:** 10.3390/ijms262311684

**Published:** 2025-12-02

**Authors:** Shengcheng Deng, Yufan Yang, Dapeng Gao, Jiajun Gao, Yuanyan Xiong

**Affiliations:** MOE Key Laboratory of Gene Function and Regulation and Guangzhou Key Laboratory of Healthy Aging Research, School of Life Sciences, Sun Yat-sen University, Guangzhou 510275, China; dengshch6@mail.sysu.edu.cn (S.D.);

**Keywords:** tumor stemness, lncRNAs, single-cell transcriptomics, pan-cancer analysis

## Abstract

Tumor stemness represents a key biological process that drives tumor progression and therapeutic resistance across various cancer types. To systematically elucidate the regulatory roles of long non-coding RNAs (lncRNAs) in this process, we integrated bulk transcriptomic data from The Cancer Genome Atlas (TCGA) with publicly available pan-cancer single-cell transcriptomic atlases. Using machine-learning-based stemness metrics, we successfully quantified stemness features and identified unique lncRNA gene sets for each cancer type at the bulk data level. The high-stemness subtype exhibited enhanced proliferation, an immunosuppressive microenvironment, and profound metabolic reprogramming. Based on these findings, we constructed a robust prognostic model with remarkable predictive performance across multiple cancer types. At the single-cell resolution, we reconstructed the dynamic trajectory of stemness evolution, uncovering distinctive metabolic and cell-communication patterns within cancer stem cells (CSCs). This multi-scale analysis consistently nominated a core set of regulatory lncRNAs, including NEAT1 and MALAT1. Our work not only nominates potential targets for stemness-directed therapy but also provides a comprehensive framework for understanding lncRNA-driven mechanisms of cancer aggressiveness and resistance.

## 1. Introduction

Malignant tumors have emerged as a major public health challenge worldwide, posing a severe threat to human health [1,2]. Their high mortality is primarily attributed to distant metastasis, post-treatment recurrence, and the development of resistance to conventional chemotherapy and radiotherapy. Metastasis remains the primary cause of approximately 90% of cancer-related deaths worldwide, and mounting evidence indicates that this lethal process is intimately connected with cancer stem cells (CSCs) [3,4]. CSCs represent a specialized subpopulation within tumors, characterized by their capacity for unlimited self-renewal and multi-directional differentiation potential [5,6]. These cells function as the core “seed” population that drives tumor initiation, maintains cellular heterogeneity, and promotes both local growth and distant metastasis [3,7,8]. The biological state of CSCs is highly dynamic and engages in continuous bidirectional crosstalk with the tumor microenvironment (TME) [7,9,10]. Under stimuli such as hypoxia or inflammatory cytokines, non-stem tumor cells can undergo dedifferentiation to reacquire stem-like properties—a hallmark of phenotypic plasticity that substantially increases the complexity of targeted therapies [11,12,13]. Moreover, CSCs are intrinsically resistant to conventional chemo- and radiotherapy due to their robust DNA damage repair machinery and active drug efflux pump systems [14,15,16]. Consequently, a deeper understanding of the molecular mechanisms governing tumor stemness is crucial for developing novel strategies to effectively suppress tumor progression and overcome therapeutic resistance.

To enable the quantitative assessment of tumor stemness, computational frameworks have been developed that leverage the molecular signatures of embryonic stem cells (ESCs) and induced pluripotent stem cells (iPSCs). Among the most influential are the mRNA expression-based (mRNAsi) and DNA methylation-based (mDNAsi) stemness indices established by Malta et al. (2018) [17]. These indices utilize machine learning models trained on pluripotent stem cell profiles to infer the degree of stem-like dedifferentiation in bulk tumor samples [18,19,20,21]. Collectively, these indices and established protein-coding gene sets—encompassing key stemness factors (e.g., OCT4, SOX2, NANOG) and critical signaling pathways (e.g., Wnt, Notch)—provide a robust toolkit for systematically evaluating tumor stemness [22,23,24,25]. This approach allows researchers to connect stemness properties with clinical outcomes, therapeutic resistance, and tumor heterogeneity, thereby offering a multidimensional perspective on CSC biology [26,27,28].

LncRNAs are regulatory RNA molecules longer than 200 nucleotides that lack protein-coding capacity [29,30,31,32]. They modulate gene expression through multiple mechanisms—such as acting as molecular scaffolds, decoys (or “sponges”), and guide molecules [33,34]—thereby influencing epigenetic states, transcriptional activity, and post-transcriptional processes [35,36,37]. Importantly, recent pan-tissue transcriptomic studies have extended this understanding, providing robust evidence for lncRNA-mediated multilayered regulation, particularly in chromatin remodeling and complex RNA metabolism [38]. Accumulating evidence has demonstrated that aberrant lncRNA expression plays a pivotal role in the maintenance and regulation of tumor stemness. Key lncRNAs, including HOTAIR [39,40], H19 [41,42], NEAT1 [43,44], LINC00511 [45,46], and FMR1-AS1 [47,48], have been recognized as central components of the regulatory hierarchy governing cancer stem-cell properties. Nevertheless, the majority of existing studies remain confined to individual lncRNAs in single cancer types, and a comprehensive, pan-cancer analysis that systematically elucidates the relationship between lncRNAs and tumor stemness is still lacking.

While transcriptome-wide analyses have already been widely applied to characterize lncRNA functions and regulatory patterns in individual cancer contexts—including comprehensive investigations in breast cancer, renal cell carcinoma, and hepatocellular carcinoma [49,50,51]—a systematic, pan-cancer landscape linking these regulators to tumor stemness remains to be fully established. To bridge this gap, our study integrates comprehensive pan-cancer transcriptomic data from TCGA with cutting-edge single-cell RNA-seq datasets. In line with emerging evidence that machine learning frameworks hold strong promise for advancing lncRNA classification and biomarker discovery [52], we develop and validate a novel pan-cancer prognostic model based on multiple lncRNAs. We systematically screened for lncRNAs significantly associated with either the mRNAsi or the mDNAsi, thereby constructing two pan-cancer applicable lncRNA gene sets related to tumor stemness. Through rigorous molecular subtyping, functional enrichment analysis, prognostic modeling, interaction-network construction, and single-cell level characterization, we comprehensively elucidated the systematic functions of these lncRNA sets in maintaining tumor stemness, shaping immunosuppressive microenvironments, regulating metabolic reprogramming, and influencing clinical outcomes. Unlike previous studies that have largely considered lncRNA regulation and machine learning in isolation, our work thoroughly explores the biological foundations and clinical applicability of the proposed model. Importantly, this study represents the first systematic characterization of tumor stemness-related lncRNA expression patterns and regulatory networks at single-cell resolution, providing new theoretical foundations and scientific evidence for understanding tumor heterogeneity and developing broad-spectrum therapeutic strategies targeting tumor stemness. The significance of this work extends beyond the mere cataloging of relevant lncRNAs; it establishes a conceptual framework for understanding how lncRNA networks coordinate complex biological processes that drive cancer progression and treatment resistance.

## 2. Results

### 2.1. A Pan-Cancer Landscape of lncRNA Signatures Associated with Tumor Stemness

Tumor stemness is a critical driver of tumor progression and therapeutic resistance. While long non-coding RNAs (lncRNAs) are emerging as key regulators of this process, a systematic understanding of their pan-cancer roles remains limited. To quantitatively assess stemness, foundational tools were established by Malta and colleagues who constructed the mRNA expression-based (mRNAsi) and DNA methylation-based (mDNAsi) stemness indices from TCGA data [17]. Building upon these foundational insights, our study employed Least Absolute LASSO regression analysis to identify lncRNAs associated with tumor stemness across the cancer transcriptome (Appendix A). Through systematic analysis of 33 distinct cancer types within the TCGA cohort, we successfully identified multiple lncRNAs exhibiting significant correlations with both mRNAsi and mDNAsi (Figure 1a, Appendix A). A particularly noteworthy finding was that these lncRNAs could function as independent predictors, effectively recapitulating the prediction results of mRNAsi and mDNAsi even in the complete absence of protein-coding genes (PCGs) expression data. This observation, substantiated by rigorous validation (Appendix A), demonstrates the remarkable accuracy and robustness of the selected lncRNA set in quantifying tumor stemness without requiring conventional protein-coding gene information.

We delineated 57 core lncRNAs consistently associated with stemness across at least seven cancer types (*p* < 0.01), revealing a conserved regulatory network (Figure 1b). A network analysis further pinpointed colorectal (COAD), gastric (STAD), and breast (BRCA) cancers as hubs of extensive lncRNA-stemness interactions, suggesting these malignancies are particularly reliant on lncRNA-mediated stemness regulation (Figure 1c). The relative importance of specific lncRNAs varied across cancer types, as detailed by their weight distributions in our model (Figure 1d), providing a roadmap for cancer-specific therapeutic targeting.

### 2.2. An lncRNA-Based Stemness Subtype System Unveils Proliferative, Immunosuppressive, and Metabolic Hallmarks of Aggressive Tumors

To evaluate the clinical utility of the identified stemness-related lncRNAs, we performed unsupervised consensus clustering on TCGA samples using lncRNA expression profiles devoid of protein-coding gene information. Through meticulous evaluation of the consistency cumulative distribution function (CDF) and Delta area plots, we determined a stable clustering solution with K = 2 across all cancer types, where subtype members demonstrated high internal consistency and maximal inter-subtype differences (Appendix A). This lncRNA-only classifier effectively recapitulated stemness differences established by prior multi-omics indices (Figure 1e,f), confirming that lncRNA profiles are potent determinants of tumor stemness. We therefore designated the resulting groups as high-stemness and low-stemness subtypes for all subsequent analyses.

In low-grade glioma (LGG), heatmap analysis vividly demonstrated that these two subtypes exhibit markedly different lncRNA expression patterns (Figure 2a,b, Appendix A). The high-stemness subtype displayed specific lncRNA modules with significant upregulation (e.g., HOMER3-AS1) or downregulation (e.g., AC017100.1), forming clear molecular subtype boundaries. This evidence strongly supports the notion that the lncRNA expression features we identified can serve as reliable molecular markers for precise classification of tumor stemness subtypes.

To further unravel the biological implications of this classification, we conducted Gene Set Variation Analysis (GSVA) on tumor samples. The resulting GSVA enrichment score heatmap clearly revealed dramatically different pathway landscapes between high- and low-stemness subtypes (Figure 2c,d). Pan-cancer statistical analysis (Appendix A) demonstrated that the high-stemness subtype consistently exhibited significant activation of pathways related to cell proliferation and genome maintenance. Key pathways including cell-cycle, DNA Replication, Homologous Recombination, and various DNA damage repair mechanisms (such as Mismatch Repair and Nucleotide Excision Repair) were consistently upregulated in the high-stemness subtype across most cancer types. This pattern suggests that high-stemness subtypes possess higher proliferation rates and enhanced DNA damage response mechanisms to maintain genomic stability—a characteristic that aligns with the known efficient DNA damage sensing and repair capabilities of CSCs. Concurrently, the mRNAsi high-stemness subpopulation displayed reduced “adhesion” and “communication” capabilities with the surrounding microenvironment, which may facilitate cellular detachment from primary sites or maintenance of an undifferentiated state.

Furthermore, this lncRNA-based subtyping revealed significantly reprogrammed immune and metabolic landscapes. Compared to the low-stemness subgroup, the high-stemness subgroup exhibited substantially reduced infiltration levels of multiple key immune cells within the tumor microenvironment (TME). This included core effector cells of adaptive immunity (such as activated CD8+ T cells and CD4+ T cells) as well as critical components of innate immunity (including natural killer cells, dendritic cells, and macrophages) (Figure 2e). Importantly, this immunosuppressive phenomenon demonstrated pan-cancer universality. Spearman correlation analysis heatmap visually demonstrated that in most cancer types, the stemness index (mRNAsi) exhibited widespread and significant negative correlations with the infiltration levels of various immune cells (Appendix A). This indicates that high-stemness tumors tend to form “immune-cold” phenotypes, potentially representing a core mechanism for evading host immune surveillance and promoting tumor progression.

At the metabolic level, the high-stemness subgroup displayed a highly active metabolic state. In LGG specifically, multiple core metabolic pathways (including the tricarboxylic acid cycle, glutamate and glutamine metabolism) showed significantly upregulated activity in the high-stemness subtype (Figure 2f). Pan-cancer analysis further confirmed that this metabolic reprogramming represents a universal feature, with the stemness index (mRNAsi) showing significant positive correlations with the activity of pathways such as Glycolysis, Pyruvate metabolism, and Metabolism of amino acids and derivatives (Appendix A). This “Warburg effect”-like metabolic remodeling provides sufficient energy and biosynthetic precursors to support the rapid proliferation of high-stemness cells. Collectively, the stemness subtype defined by lncRNA expression profiles serves not only as an effective prognostic stratification tool but also profoundly reveals the intrinsic connections between tumor stemness and three core malignant features: highly proliferative capacity, immune evasion mechanisms, and metabolic reprogramming.

### 2.3. Stemness-Associated lncRNA Subtype System Stratifies Prognosis Across Cancers

To investigate the clinical prognostic value of stemness-related lncRNAs, we constructed a risk scoring model based on lncRNA weights determined through Cox regression analysis (Appendix A). Using LGG as a representative example, we demonstrated the comprehensive construction process and predictive efficacy of the model. We developed two independent models derived from mRNAsi- and mDNAsi-related lncRNAs via Cox regression (Figure 3b,e). Both models demonstrated exceptional accuracy in predicting patient survival at 1, 3, and 5 years, as evidenced by time-dependent ROC curve analysis (Figure 3c,f). To verify the robustness and clinical applicability of our models, we performed independent external validation using the CGGA dataset [53,54,55,56,57,58]. Patients in the CGGA cohort were stratified into high-risk and low-risk groups based on the median risk score calculated using the formula derived from the training set. Kaplan–Meier survival analysis results confirmed significant differences in overall survival between the two groups in the external validation cohort (*p* < 0.001), with the high-risk group demonstrating substantially worse prognosis (Figure 3g,h). This compelling evidence demonstrates that our constructed lncRNA prognostic model can effectively stratify LGG patients by risk, possessing significant clinical application potential.

Prognostic lncRNAs were systematically identified across the most TCGA cancer types, with particularly abundant signatures in breast cancer (BRCA), kidney renal clear cell carcinoma (KIRC), and lung adenocarcinoma (LUAD). The consistent presence of these lncRNAs underscores that the molecular machinery of tumor stemness, as captured by our lncRNA signatures, is a fundamental driver of patient mortality across diverse cancer lineages.

### 2.4. Stemness-Associated lncRNA Networks Converge on Core Oncogenic Pathways and Therapeutic Vulnerabilities

lncRNAs construct complex and precise interaction networks with intracellular biomacromolecules through diverse mechanisms, functioning as guide molecules, decoy molecules, scaffold molecules, and molecular chaperones. To deeply investigate the molecular functions of these core stemness-related lncRNAs, we systematically analyzed their interacting protein networks using LGG from TCGA as a model system. By leveraging interactome information from the RNAInter v4.0 database, we discovered that the interactome of mRNAsi-related lncRNAs was significantly enriched in key biological processes including “signal transduction in response to DNA damage,” “stem cell differentiation,” and “ERBB signaling pathway” (Figure 4a). Kyoto Encyclopedia of Genes and Genomes (KEGG) pathway analysis further confirmed that these proteins were closely associated with core cancer pathways such as “Glioma,” “PI3K-Akt signaling pathway,” and “EGFR tyrosine kinase inhibitor resistance” (Figure 4b).

Similarly, the interactome of mDNAsi-related lncRNAs displayed a comparable functional profile, with interacting proteins significantly enriched in pathways such as “DNA damage checkpoint signaling” and “mTOR/ErbB signaling pathway” (Figure 4c,d). The common enrichment of these two lncRNA interaction networks in key pathways including “Glioma” and “PI3K-Akt/mTOR” suggests that they may drive malignant progression in LGG through coordinated regulation of similar molecular networks. This observation indicates potential redundancy and robustness in the regulatory mechanisms governing tumor stemness.

Pan-cancer analysis confirmed the universal role of this regulatory paradigm. The lncRNA interactomes consistently governed core hallmarks of cancer—genomic stability, cell-cycle progression, and cell fate commitment—across diverse tumor types (Appendix A). Moreover, they were pervasively associated with key therapeutic resistance pathways, including PD-1/PD-L1 immune checkpoint signaling and EGFR tyrosine kinase inhibitor resistance (Appendix A).

KEGG pathway analysis further confirmed that multiple classical oncogenic pathways were consistently activated across various tumors, including “PI3K-Akt,” “mTOR,” “Ras,” and “p53 signaling pathways.” Notably, these networks were also closely associated with treatment resistance mechanisms such as “EGFR tyrosine kinase inhibitor resistance” and “PD-L1 expression and PD-1 checkpoint pathway.” Collectively, the pan-cancer functional analysis powerfully demonstrates that stemness-related lncRNAs play broad and conserved critical roles in the malignant progression of multiple cancers by interacting with core proteins that regulate cell proliferation, maintain genomic stability, mediate key oncogenic signals, and influence treatment resistance.

Crucially, we translated these molecular insights into tangible therapeutic opportunities. By integrating a large-scale drug sensitivity database containing pharmacological and CRISPR screening data [59], we systematically revealed multiple potential “lncRNA–protein–drug” sensitivity association axes (Figure 4e,f, Appendix A). For instance, in the mRNAsi-related regulatory network, we identified that lncRNAs such as TMPO could interact with key proteins like EGFR, TP53, and RICTOR, thereby associating with various targeted drugs. Similarly, in the mDNAsi-related network, hub lncRNAs such as AC124067.2 and HOTAIR showed significant interactions with RICTOR, suggesting potential applications of PI3K/AKT/mTOR pathway inhibitors. These discoveries not only deepen our understanding of stemness-related lncRNA regulatory networks but, more importantly, provide new theoretical foundations and candidate strategies for precision treatment targeting high tumor stemness subpopulations. Heatmaps visually demonstrated the distribution of potential drugs related to mDNAsi and mRNAsi across different cancers (Appendix A), offering valuable insights for developing cancer-specific therapeutic approaches.

### 2.5. Single-Cell Analysis Reveals Tumor Stemness Differences and Unveils Their Dynamic Evolutionary Trajectories

To investigate tumor stemness characteristics and internal heterogeneity at single-cell resolution, we integrated a comprehensive pan-cancer single-cell transcriptomic atlas (GSE210347), encompassing 10 diverse cancer types including bladder cancer (BLCA), breast cancer (BRCA), and colorectal adenocarcinoma (COAD). Using BRCA as an illustrative example, we first performed unsupervised annotation of all cells through UMAP dimensionality reduction (Figure 5a). To precisely identify malignant tumor cells, we conducted copy number variation (CNV) analysis on single cells using both inferCNV and CopyKat algorithms (Appendix A), integrating the inference results from both methods to accurately define the malignant cell population (Figure 5b). Based on this rigorous identification process, we employed a combinatorial strategy to robustly quantify tumor stemness. While the mRNAsi score was originally developed for bulk sequencing, its applicability to single-cell transcriptomics was explicitly demonstrated in the original study, using scRNA-seq datasets and has been widely adopted in recent high-impact studies to define malignant subpopulations [17,60,61]. Complementarily, we utilized CytoTRACE to measure differentiation potential, which serves as a validated intrinsic proxy for “oncogenic dedifferentiation” and stem-like traits. Notably, the validity of this approach in identifying high-potential tumor cells is well-documented across various cancers [62,63,64,65]. Consequently, we adopted an intersection strategy to ensure high confidence, defining CSCs as cells ranking in the top 20% of both CytoTRACE and mRNAsi scores. Notably, cancer cell clusters exhibited higher CytoTRACE stemness indices (Figure 5c), with similar trends observed consistently across other cancer types (Figure 5d), confirming the universal relevance of stemness properties in tumor biology.

To further elucidate the dynamic changes in stemness within cancer cell populations, we performed pseudotime analysis on cancer cell subpopulations in BLCA using Monocle3. We established the cell cluster with the highest CytoTRACE score as the starting point of the trajectory (Figure 5e). Along this inferred differentiation path, we observed that two different stemness scoring indicators, CytoTRACE and mRNAsi, exhibited highly consistent downward trends (Figure 5f,g). This finding clearly indicates the existence of a continuous dynamic process within tumors, evolving from high-stemness/low-differentiation states toward low-stemness/high-differentiation states. This observation provides direct evidence for the phenotypic plasticity of tumor cells and their capacity to transition between stem-like and differentiated states.

This critical finding reveals that stemness is not a monolithic state but a multi-dimensional property, whose manifestation and measurement depend profoundly on cancer-type-specific biology. This complexity necessitates context-aware definitions of stemness for accurate biological interpretation and therapeutic targeting.

### 2.6. Metabolic Reprogramming and Dominant Cell Communication Roles of Cancer Stem Cells

We identified cancer stem cells (CSCs) across the pan-cancer single-cell atlas as cell clusters with both high CytoTRACE scores and substantial abundance (>5% of tumor cells; Figure 6a). Using BLCA as a representative example, AUCell analysis showed that compared to ordinary tumor cells, CSCs exhibited significantly enhanced epithelial–mesenchymal transition (EMT) and cell-cycle pathway activities (Figure 6b), indicating that CSCs not only possess stronger proliferative capacity but also demonstrate higher cellular plasticity and migration/invasion potential. These findings align with the established properties of CSCs as drivers of tumor progression and metastasis.

Further metabolic pathway analysis revealed that CSCs exhibit extensive metabolic reprogramming characteristics. Gene Set Enrichment Analysis (GSEA) showed that pathways closely related to biosynthesis and proliferation were significantly upregulated in CSCs compared to ordinary tumor cells, including D-glutamine and D-glutamate metabolism, arginine biosynthesis, and folate-mediated one-carbon metabolism pathways. Conversely, a series of catabolic pathways were significantly downregulated, particularly branched-chain amino acid (valine, leucine, and isoleucine) degradation, ketone body metabolism, and various lipid metabolism pathways (Figure 6c). This metabolic profile suggests a strategic adaptation where CSCs prioritize biosynthetic processes while suppressing catabolic pathways. This metabolic suppression may represent a key mechanism for maintaining stemness, by avoiding the production of signaling molecules that induce differentiation (such as branched-chain amino acid metabolites that activate mTORC1 or ketone bodies that alter histone acetylation), thereby locking cells in an undifferentiated, highly plastic state.

Pan-cancer analysis confirmed that pathways such as steroid hormone biosynthesis, D-arginine and D-ornithine metabolism, and glycosaminoglycan biosynthesis consistently showed downregulation trends across most cancer types, suggesting that these metabolic changes may represent conserved stemness maintenance features across diverse cancer contexts (Appendix A). This observation highlights the potential for developing metabolic interventions that target CSCs across multiple cancer types.

CSCs actively reshape their microenvironment to promote their own survival, evade immune attacks, and induce angiogenesis. Our analysis revealed that CSCs exhibit high frequency and intensity of cell–cell communication. (Figure 6d,e). Compared to ordinary cancer cells, CSCs demonstrated stronger interaction intensity and higher interaction frequency with various cell types including tumor cells themselves, endothelial cells, and smooth muscle cells (Appendix A). Pan-cancer scale analysis showed that CSCs exhibited significantly different communication activity from ordinary cancer cells in the vast majority of cancer types (Appendix A), underscoring the universal importance of cell communication in CSC biology.

In BLCA specifically, CSCs exhibited stronger activity in multiple pathways whether acting as signal senders or receivers (Figure 6f), particularly in collagen (COLLAGEN) and laminin (LAMININ) signaling pathways (Figure 6g,h). When CSCs function as signal receivers, various collagens interact with integrin receptors on the CSC surface, and this continuous incoming signal appears crucial for maintaining the self-renewal capacity of CSCs. The enhanced MIF-CD74_CD44 signaling pathway suggests that CSCs may utilize this mechanism to promote their own proliferation and create an immunosuppressive microenvironment (Appendix A). When acting as signal sources, CSCs actively secrete laminin to remodel the extracellular matrix (ECM), constructing a microenvironment favorable for their own survival and invasion, while also driving angiogenesis by secreting CXC chemokines and pleiotrophin (MDK), providing essential support for tumor growth (Appendix A). These findings collectively indicate that CSCs actively regulate the TME by dominating cell communication centered on ECM components, which likely represents a key mechanism for maintaining stemness, promoting angiogenesis, and facilitating tumor invasion and metastasis.

### 2.7. Identifying Core lncRNA Regulons That Orchestrate Stemness at Single-Cell Resolution

To precisely identify lncRNAs regulating stemness at the single-cell level, we first employed a metacell construction strategy to overcome the inherent sparsity of single-cell transcriptomic data. Using BLCA as a representative example, we successfully screened a batch of key lncRNAs highly correlated with both CytoTRACE and mRNAsi stemness scores using the LASSO regression model (Figure 7a,b), with the robustness of the model systematically evaluated through rigorous validation procedures (Appendix A). Analysis results showed that HOTAIRM1, SBF2-AS1, and LINC00958 were the lncRNAs most significantly associated with CytoTRACE scores; whereas PVT1, EMX2OS, and TMPO-AS1 were most significantly associated with mRNAsi scores. Heatmaps visually demonstrated the expression patterns of the top 15 lncRNAs most related to these two stemness scores in BLCA, clearly showing that their expression levels were highly correlated with tumor stemness status (Appendix A).

To further elucidate the regulatory mechanisms of these lncRNAs, we constructed lncRNA-target gene regulatory networks using the GENIE3 algorithm, revealing lncRNA-centered regulatory networks (Figure 7c,d). The expression scores of downstream regulated proteins showed close correlation with stemness scores (Figure 7e,f). From these networks, we identified regulon modules composed of core lncRNAs and their target genes. Activity analysis showed that the overall activity of top lncRNA regulons exhibited high correlation with cellular stemness scores (Figure 7e,f), further confirming the key roles of these lncRNAs in regulating stemness states. lncRNA target genes related to CytoTRACE scores were mainly enriched in processes such as “DNA replication initiation,” “DNA unwinding,” and “cell-cycle DNA replication” (Figure 7g), while target genes related to mRNAsi scores were significantly enriched in pathways such as “nuclear division,” “chromosome segregation,” and “regulation of cell-cycle phase transition” (Figure 7h). This functional enrichment pattern aligns perfectly with the high proliferative characteristics exhibited by high-stemness tumor cells, providing mechanistic insights into how lncRNAs regulate tumor stemness.

To identify core regulatory factors with pan-cancer applicability, we systematically analyzed the occurrence frequency and regulatory weights of stemness-related lncRNAs across 10 cancer types. This comprehensive analysis identified a batch of core lncRNAs that exhibited strong associations and high regulatory potential across multiple cancers, including NEAT1, MALAT1, GAS5, TP53TG1, and LINC00941 (Figure 7i), many of which have been previously confirmed to be closely related to cancer progression. To verify the consistency of research results across different data scales, we systematically compared stemness-related lncRNAs identified by both TCGA bulk data and single-cell data. Results showed that there was a significant intersection between the lncRNA gene sets identified by the two platforms (Appendix A). This cross-platform, cross-scale co-validation result powerfully demonstrates that the identified lncRNA gene set plays a stable and core role in regulating tumor stemness, thereby providing potential molecular targets and solid theoretical foundations for diagnosis and treatment targeting tumor stemness.

## 3. Discussion

Tumor stemness represents a fundamental biological process that drives tumor heterogeneity, metastasis, recurrence, and therapeutic resistance across diverse cancer types. Although lncRNAs as key regulatory molecules have received considerable attention in recent years, previous studies have revealed the function of lncRNAs in modulating stemness in specific tumor types [50,65]. Our work extends this foundation through a systematic, multi-omics investigation that integrates TCGA pan-cancer cohorts with single-cell transcriptomic datasets. This strategy enables a novel, cross-cancer characterization of lncRNA-mediated stemness.

A core innovation of our work lies in the successful construction of two lncRNA expression feature sets using advanced machine learning methods that operate independently of PCGs. Previous transcriptome-based studies have not established an lncRNA-only machine-learning system for pan-cancer stemness evaluation [49], highlighting the methodological novelty of our framework. This finding has significant implications for clinical practice, as it suggests that lncRNA expression profiles alone may be sufficient for evaluating tumor stemness without requiring comprehensive protein-coding gene expression data. These feature sets not only precisely stratify tumors into high- and low-stemness subtypes but also reveal the malignant phenotypic characteristics of high-stemness subtypes, which include enhanced proliferation, activated DNA repair mechanisms, and the formation of “immune-cold” microenvironments. Our results not only confirm prior studies on the proliferative and immunosuppressive nature of aggressive cancers [66,67] but also provide a definitive, lncRNA-mediated mechanistic link between these phenotypes and the dysregulation of tumor stemness across cancer types.

The pan-cancer prognostic model constructed based on these lncRNA features demonstrates powerful risk stratification capabilities across multiple cancer types. The most significant translational value of our study emerges from the construction of the “lncRNA–protein–drug” intervention pathways, consistent with pathway-centric findings in earlier single-cancer studies [50,66]. This molecular insight not only explains the biological basis of the model’s prognostic prediction capability but also suggests a series of potential “lncRNA–protein–drug” intervention pathways. By constructing these pan-cancer “lncRNA–protein–drug” regulatory axes, our work extends previous findings on lncRNA-mediated oncogenic pathways [51,68] and reveals significant translational potential for hard-to-treat cancers.

Beyond prior examinations of single-cell stemness dynamics, our study provides mechanistic validation through a systematic modeling framework that integrates lncRNA regulation at a pan-cancer scale [69,70]. By defining cell clusters with the highest CytoTRACE scores as CSCs, we uncovered their unique metabolic reprogramming characteristics: upregulation of biosynthetic pathways coupled with downregulation of catabolic pathways (such as branched-chain amino acid degradation and ketone body metabolism). This “biosynthesis-intensive but catabolism-stagnant” metabolic pattern may represent a fundamental mechanism for maintaining stemness by avoiding the production of differentiation-inducing metabolites. This novel hypothesis provides a metabolic perspective on stemness maintenance that complements existing molecular and epigenetic explanations.

Furthermore, our cell communication analysis revealed that CSCs are not passive entities but active “shapers” of the TME. In light of known metabolic and communication alterations in CSCs [71],we systematically integrate single-cell data to thereby associate these phenotypes with precise lncRNA-defined stemness states. In bladder cancer, they actively communicate with surrounding cells through enhanced extracellular matrix-related signals such as collagen and laminin, as well as immune regulatory signals like MIF, constructing ecological niches favorable for their survival and immune exemption.

The greatest strength of our study lies in its cross-platform, cross-scale design. Having established the conceptual role of lncRNAs in stemness regulation [69], we leverage systematic multi-omics and single-cell analyses across a spectrum of tumors to provide mechanistic and translational validation. These lncRNAs represent key regulatory factors stably present across different data dimensions and biological scales, with hub lncRNAs such as NEAT1 and MALAT1 emerging as particularly significant players. Additionally, by constructing “lncRNA–protein–drug” association axes, we linked the lncRNA regulatory network with known targeted therapies (such as EGFR TKIs and mTOR inhibitors) and immunotherapy resistance mechanisms, providing concrete pathways for developing combination therapeutic strategies targeting CSCs.

Despite these significant advances, our study has certain limitations. First, our findings are primarily based on bioinformatics analysis; although reliability is enhanced through multi-dataset and cross-platform (bulk and single-cell) validation, the specific molecular mechanisms, such as how key lncRNAs precisely regulate downstream target genes, still need to be ultimately confirmed through rigorous in vitro and in vivo functional studies. Second, while we employed strict regularization (λ1se) to prevent overfitting, models derived from small cohorts (e.g., DLBC, CHOL) should be interpreted with caution due to inherent statistical limitations and necessitate further external validation. Third, while we integrated extensive pan-cancer data, the inherent heterogeneity of cancer means that unique lncRNA regulatory mechanisms in specific cancer types may still require further exploration. Finally, our current framework is strictly confined to the analysis of lncRNAs associated with tumor stemness. We have not yet established specific lncRNA scoring metrics for other critical phenotypes, such as metabolic pathways and immune microenvironments. Future work will focus on developing broader computational tools to quantify lncRNA signatures for these distinct biological processes and validating the proposed “lncRNA–protein–drug” axes through experimental approaches.

In summary, this study systematically characterized the core role of lncRNAs in regulating pan-cancer stemness through multi-omics integrative analysis. Extending beyond cancer-type–specific descriptions of related phenomena [51,68],our integrative analysis provides a unified, pan-cancer perspective on stemness regulation, centered on lncRNAs and validated through single-cell profiling. Our primary finding is the construction of the “lncRNA–protein–drug” regulatory axes, which not only explains the biological basis of the model’s prediction capability but also provides a solid theoretical foundation for developing novel precision therapeutic strategies targeting tumor stemness and resistance by targeting the lncRNA regulatory network.

## 4. Materials and Methods

### 4.1. Data Collection

DNA-methylation-based stemness index (mDNAsi) and mRNA-expression-based stemness index (mRNAsi) for TCGA datasets were obtained from previous studies [17], in which these indices were calculated using one-class logistic regression (OCLR).

Bulk RNA sequencing (RNA-seq) data were downloaded from The Cancer Genome Atlas (TCGA) through the Genomic Data Commons (GDC) data portal (https://portal.gdc.cancer.gov/), covering 10,852 samples across 33 cancer types (Appendix A). Clinical data, including survival outcomes, were also obtained from the same source using the TCGAbiolinks (v2.35.3) [72,73,74] and importing into R (v4.3.1, http://www.r-project.org) [75].

Single-cell RNA-seq (scRNA-seq) data were derived from Gene Expression Omnibus (GEO) under accession number GSE210347 [76], comprising 232 single cell transcriptome samples (normal = 31; adjacent = 54; tumor = 148). After rigorous quality control, a total of 643,234 cells were retained for downstream analysis. The detailed cell distribution across the 10 cancer types (e.g., BRCA: 41041 cells; LUAD: 168556 cells) is provided in Appendix A.

### 4.2. Stemness Signature Construction with LASSO Regression

To identify stemness-associated lncRNAs for each of the 33 cancer types in the TCGA cohort, we performed individualized least absolute shrinkage and selection operator (LASSO) regression using the glmnet (v4.1.10) [77,78,79] package in R, modeling the relationship between the expression profiles of 19,374 long non-coding RNAs and the corresponding pan-cancer stemness indices (mDNAsi and mRNAsi) as previously described. Notably, these lncRNAs were annotated based on GENCODE v36, with pseudogenes explicitly excluded to ensure high specificity [80]. Specifically, for each cancer type, we fitted a separate LASSO regression model with lambda (λ) optimized via 10-fold cross-validation to minimize the mean squared error (MSE). The model selected an optimal λ value for each cancer type (within the range of log(λ) ≈ −6 to −4), which imposed a stringent penalty to prevent overfitting, effectively shrinking the coefficients of non-informative lncRNAs to zero while retaining only those most strongly associated with the stemness indices.

### 4.3. LncRNA-Cancer Project Co-Expression Network Analysis

An lncRNA-cancer project co-expression network was then constructed to map associations between stemness-related lncRNAs and tumor types using the igraph (v2.1.4) [81] package in R. The network included lncRNA nodes and cancer project nodes, with edges weighted by significant Spearman correlations (FDR < 0.05).

### 4.4. Stemness Group Stratification

To elucidate the heterogeneity of stemness across samples, we performed consensus clustering analysis separately for each of the 33 cancer types with the ConsensusClusterPlus (v1.66.0) [82] package in R. This unsupervised machine learning approach was applied to the expression profiles of stemness-associated lncRNAs identified previously. The analysis was configured with 1000 iterations to ensure the stability of the classifications, and the cluster count (k) was set to 2 for all cancer types to stratify patients into two fundamental subgroups with high and low stemness, respectively. The partitioning was conducted using the k-means algorithm, with Pearson correlation employed as the distance metric to assess the similarity between samples. The optimal number of clusters was confirmed by analyzing the cumulative distribution function (CDF) curve.

### 4.5. Gene Set Variation Analysis

Single-sample gene set enrichment analysis (ssGSEA) was implemented using GSVA (v1.50.5) [83] to comprehensively characterize the functional states associated with stemness, including the immune infiltration score and metabolism activity levels, between the high- and low-stemness groups across multiple cancer types.

The differentially expressed genes (DEGs) between the high-stemness group and the low-stemness group were identified using the limma (v3.50.1) [84] package in R. A false discovery rate (FDR) threshold of <0.05 and an absolute log2 fold change of >1 were applied to define statistically significant and biologically relevant DEGs. Subsequently, gene set enrichment analysis (GSEA) was performed using the GSVA package to identify Gene Ontology (GO) biological processes and Kyoto Encyclopedia of Genes and Genomes (KEGG) pathways that were significantly enriched in high-stemness versus low-stemness groups. The GSEA was conducted using a pre-ranked list of genes based on their expression fold changes, with 1000 permutations to assess statistical significance. Pathways with a normalized enrichment score (NES) > 1.5 and FDR < 0.25 were considered significantly enriched.

### 4.6. Prognostic Signature Construction with LASSO and Cox Regression

Univariate Cox regression analysis, performed with the survival (v3.8.3) [85,86] package in R, identified multiple lncRNAs in stemness signature that were significantly associated with patient prognosis (*p* < 0.05) across the TCGA cohort. The LASSO regression was applied using the glmnet package to refine the prognostic signature. Several key lncRNAs retained by LASSO regression were then incorporated into a multivariate Cox proportional hazards model to construct the final prognostic signature. The risk score for each patient was calculated as a linear combination of the expression levels of the selected lncRNAs, weighted by their respective coefficients from the multivariate Cox model: Risk score = (β_1_ × expr_1_) + (β_2_ × expr_2_) + … + (βₙ × exprₙ). In this equation, each coefficient (β) represents the log hazard ratio for its corresponding lncRNA, where a positive value indicates an association with poorer survival. Based on the median risk score, patients were stratified into high- and low-risk groups within each cancer type.

To validate the performance of our model, we employed Kaplan–Meier survival analysis with log-rank tests using the survminer (v0.5.0) [87] package in R. Additionally, receiver operating characteristic (ROC) analysis performed using the survival package confirmed the model’s robust predictive accuracy by yielding area under the curve (AUC) values above 0.7 at 1, 3, and 5 years of follow-up.

### 4.7. Interaction Network Analysis of the Prognostic Signature

Functional annotation of the interactomes for the prognostic signature lncRNAs [88] was first conducted using clusterProfiler (v4.10.1) [89] for KEGG and GO analyses, with significantly enriched terms identified using a false discovery rate (FDR) threshold of <0.05. To construct the lncRNA–protein–drug interaction axis, we subsequently integrated our prognostic signature with published drug sensitivity data [59] by linking the co-expressed protein partners of the lncRNAs to their corresponding therapeutic compounds and visualized these associations as a Sankey diagram using the networkD3 (v0.4.1) [90] package in R.

### 4.8. Single-Cell Data Processing

We performed an integrated analysis on 232 single-cell transcriptome samples from 10 cancer types (GSE210347) [76]. Batch correction and data integration were performed using the harmony (v1.2.3) [91] package in R to mitigate technical variations across samples. Cell clustering was conducted with the Seurat (v5.3.0) [92,93,94,95] package applying a graph-based clustering algorithm (resolution = 0.5), followed by uniform manifold approximation and projection (UMAP) visualization using the top 30 principal components. Cell type annotation was automatically assigned with SingleR (v2.4.1) [96] by referencing the celldex (v1.12.0) index [96].

### 4.9. Cancer Stem Cell Characterization

To identify tumor cells, we performed copy number variation (CNV) analysis on the integrated single-cell transcriptome data. Two approaches were used: the copykat (v1.1.0) [97] package for expression intensity-based inference of large-scale chromosomal alterations, and the inferCNV (v1.24.0) [98] package for reference-based identification of somatic CNV events. Cells exhibiting significant aneuploidy or focal amplifications/deletions across chromosomes were classified as tumor cells, forming the basis for subsequent cancer stem cell (CSC) analysis.

Individual single-cell mRNAsi values were computed by applying stemness signature weights using the R package stats (v4.3.1). For validation, parallel stemness scoring was conducted with CytoTRACE2 (v1.1.0) [99], a established framework for single-cell stemness evaluation, which predicts cellular stemness based on transcriptional diversity to investigate the differentiation hierarchy within the cancer cell population, pseudotemporal trajectories were reconstructed using Monocle3 (v1.3.7) [100,101,102], with the trajectory rooted in the subclusters exhibiting the highest CytoTRACE2 scores.

Based on the integrated evidence above, we defined tumor cells that ranked within the top 20th percentile of both the lncRNA-based mRNAsi and CytoTRACE2 scores as CSCs.

### 4.10. Functional Profiling of CSC

To systematically compare the functional states between CSCs and non-CSC cancer cells, we performed a multi-faceted analysis the using AUCell (v1.24.0) [103,104] package. This tool quantified the activity of specific gene sets, including the epithelial–mesenchymal transition (EMT) signature, cell-cycle phase-specific genes and KEGG metabolic pathways, by calculating the AUC for the top 10% of expressed genes per cell. CellChat (v1.6.1) [105,106] was then employed to decipher differentially expressed ligand-receptor interactions and identify signaling pathways with significantly altered communication probability (permutation test, *p* < 0.05) between the two cell clusters.

SuperCell Construction: To mitigate data sparsity and enhance biological signals while retaining cellular heterogeneity, cancer cells were aggregated into SuperCells (approximately 50–100 cells per SuperCell) based on their cluster identities.

We then performed LASSO regression (λ = lambda.1se) on the SuperCell expression matrix using the glmnet package to identify lncRNAs associated with single-cell tumor stemness. The downstream regulatory protein networks of these stemness-associated lncRNAs were inferred using the GENIE3 (v1.24.0) [107], retaining interactions with importance scores > 0.01. Functional annotation of these proteins was carried out via GO clustering. Finally, the single-cell expression activity of this downstream regulatory proteome was quantified using the AUCell package and demonstrated a strong correlation with the tumor stemness score.

## 5. Conclusions

This comprehensive study successfully identified two lncRNA feature sets capable of independently quantifying tumor stemness through the integration of multi-omics data across diverse cancer types. Molecular subtyping based on these feature sets revealed that high-stemness tumors consistently exhibit characteristic features of active proliferation, enhanced DNA repair capacity, and immunosuppressive microenvironments. We further constructed a pan-cancer prognostic model with excellent predictive performance that demonstrated robust risk stratification capabilities across multiple cancer types. Additionally, by deciphering the intricate interaction networks of these lncRNAs, we revealed a series of potential “lncRNA–protein–drug” association axes, providing specific candidate drug strategies for targeting high-stemness tumor subpopulations with precision.

## Figures and Tables

**Figure 1 ijms-26-11684-f001:**
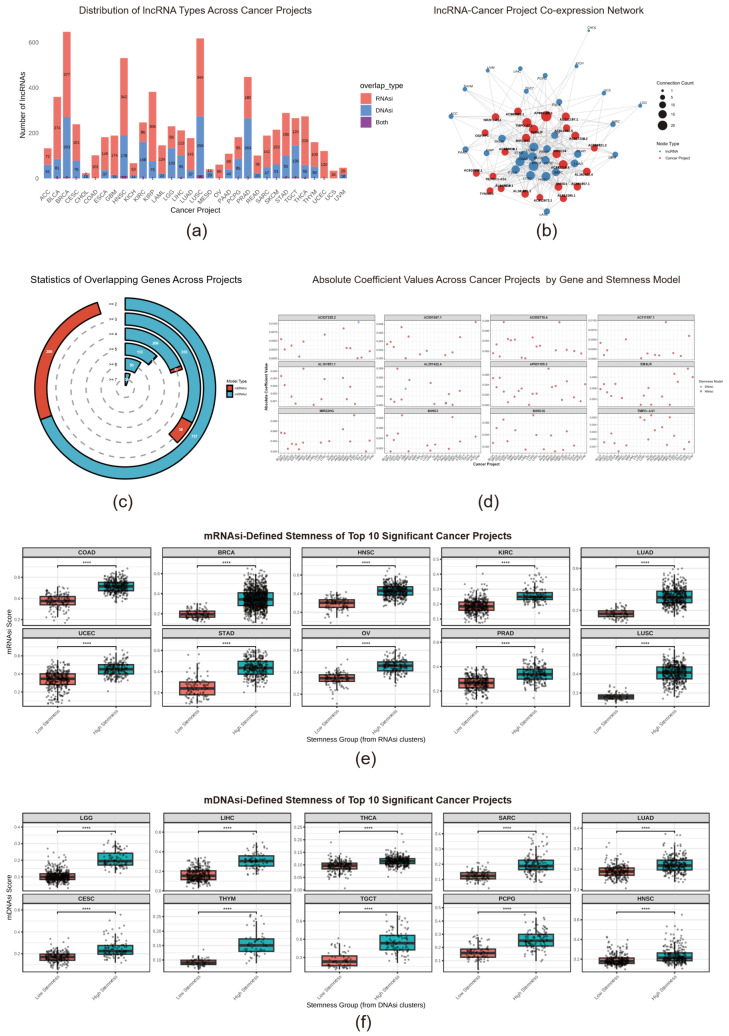
Pan-cancer Analysis of Tumor Stemness-Related lncRNAs. (**a**) Distribution of stemness-related lncRNAs across the pan-cancer landscape. The bar chart displays the number of lncRNAs associated with mRNAsi (orange-red) and mDNAsi (blue), respectively, as identified by LASSO regression across 32 TCGA cancer types. (**b**) Pan-cancer conservation analysis of core conserved lncRNAs. Blue represents lncRNAs associated with the mRNAsi model, while orange-red represents those associated with the mDNAsi model (*p* < 0.01). (**c**) The lncRNA-cancer association network. Red nodes represent lncRNAs, and blue nodes represent cancer projects. The presence of an edge indicates that the lncRNA is significantly associated with stemness in the corresponding cancer type. Node size is proportional to its connectivity. (**d**) Absolute coefficient values of hub lncRNAs in LASSO models. The dot plot shows the absolute coefficient values (weights) of hub lncRNAs in different cancer types, derived from the LASSO regression models for mRNAsi (red) and mDNAsi (blue). (**e**,**f**). Stratification of stemness indices based on lncRNA expression clusters. Consensus clustering on the expression profiles of stemness-related lncRNAs was used to classify samples for each cancer type into low-stemness (red) and high-stemness (blue) subtypes. Significant differences were observed for both (**e**) the mRNAsi and (**f**) mDNAsi. The top 10 cancer types with the most significant inter-group differences are shown. Asterisks indicate the level of statistical significance (**** *p* < 0.0001).

**Figure 2 ijms-26-11684-f002:**
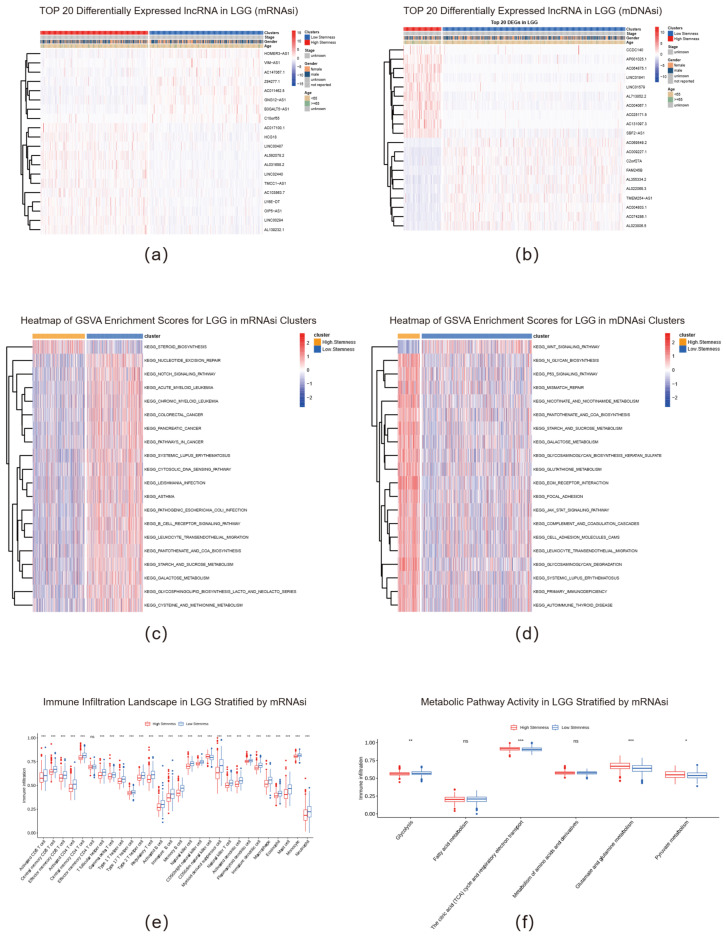
Multi-omics characteristics of stemness subtypes defined by lncRNA expression profiles. (**a**,**b**) Heatmaps showing the top 20 most differentially expressed lncRNAs between stemness subtypes defined by mRNAsi (**a**) and mDNAsi (**b**) in LGG. (**c**,**d**) GSVA enrichment score heatmaps of differentially activated KEGG pathways in subtypes defined by mRNAsi (**c**) and mDNAsi (**d**) in LGG. (**e**) Differential immune cell infiltration levels between mRNAsi-defined stemness subtypes in LGG. Boxplots display the infiltration scores of various immune cells in high- and low-stemness subtypes. (**f**) Differences in the activity of key metabolic pathways between mRNAsi-defined stemness subtypes in LGG. Boxplots show the GSVA enrichment scores for different metabolic pathways in high- and low-stemness subtypes. Asterisks indicate the level of statistical significance (***: *p* < 0.001; **: *p* < 0.01; *: *p* < 0.05; ns: not significant).

**Figure 3 ijms-26-11684-f003:**
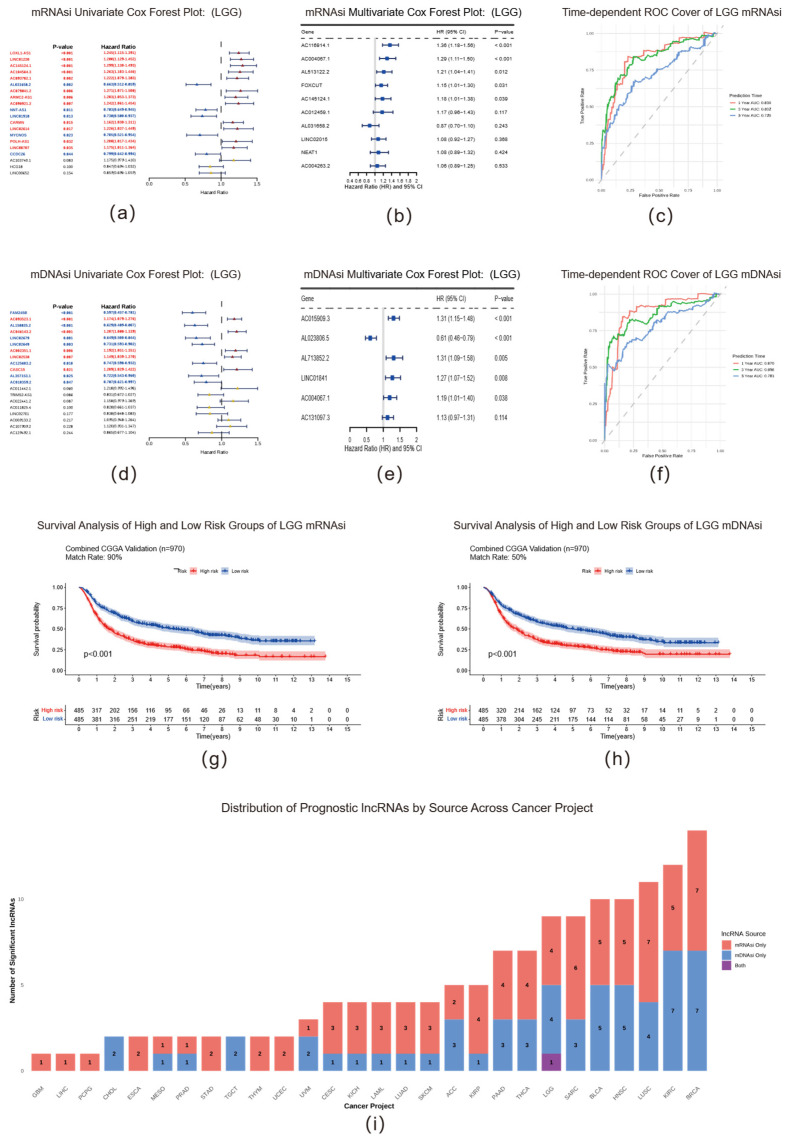
Construction and validation of a prognostic risk model based on stemness-related lncRNAs. (**a**,**b**) Forest plots of (**a**) univariate and (**b**) multivariate Cox regression analyses used to build the prognostic model based on mRNAsi-related lncRNAs in LGG. (**c**) Time-dependent ROC curves at 1, 3, and 5 years, assessing the predictive efficacy of the mRNAsi-based risk model. (**d**,**e**) Forest plots of (**d**) univariate and (**e**) multivariate Cox regression analyses for the prognostic model based on mDNAsi-related lncRNAs in LGG. (**f**) Time-dependent ROC curves for the mDNAsi-based risk model. (**g**,**h**) Independent external validation of the prognostic models in the CGGA dataset. Kaplan–Meier survival analyses of high- and low-risk groups of LGG patients in the CGGA cohort, (**g**) mRNAsi-based risk model and (**h**) mDNAsi-based risk model. *p*-values were calculated using the log-rank test. (**i**) Pan-cancer distribution of prognosis-related lncRNAs. The bar chart illustrates the number of survival-associated lncRNAs identified across cancer types, categorized by their association with mRNAsi, mDNAsi, or both.

**Figure 4 ijms-26-11684-f004:**
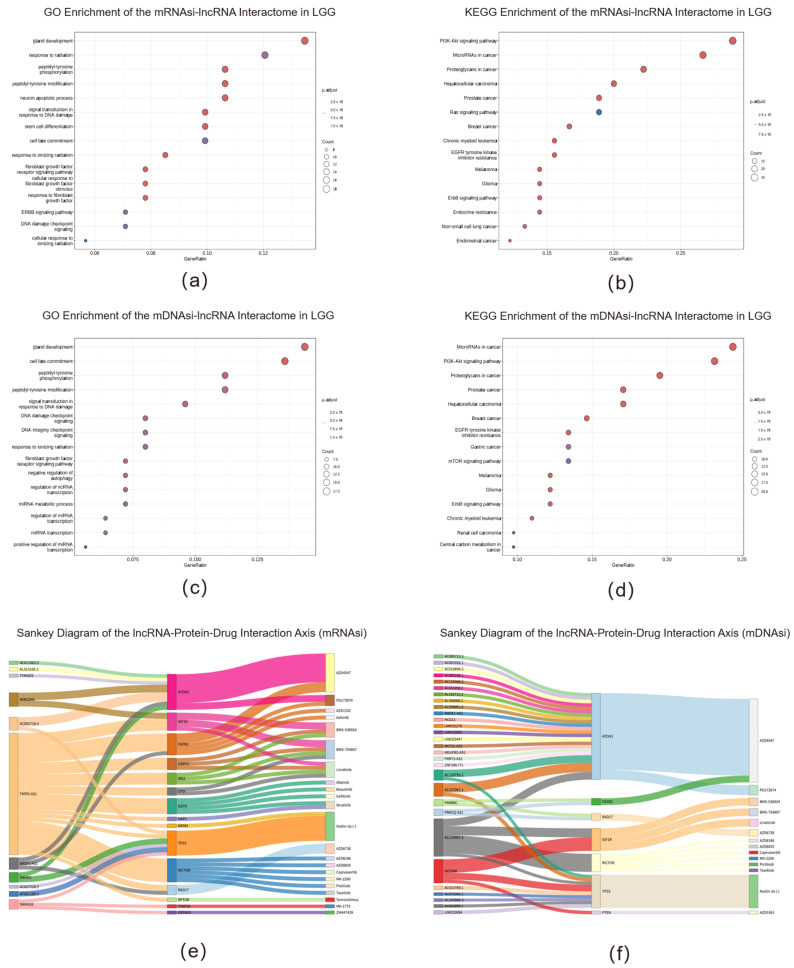
Analysis of Stemness-Related lncRNA Interacting Proteins and Drug Sensitivity. (**a**) GO enrichment analysis of the mRNAsi-lncRNA interactome in LGG. (**b**) KEGG pathway enrichment analysis of the mRNAsi-lncRNA interactome in LGG. (**c**) GO enrichment analysis of the mDNAsi-lncRNA interactome in LGG. (**d**) KEGG pathway enrichment analysis of the mDNAsi-lncRNA interactome in LGG. the color of the bars (or dots) represents the statistical significance of the enrichment, often indicated by the adjusted *p*-value, where a darker color indicates a smaller (more significant) *p*-value. The length of the bars represents the Gene Count. (**e**,**f**) Sankey diagrams illustrating potential lncRNA–protein–drug interaction axes. These diagrams show the flow from core lncRNAs (**left**) to their interacting proteins (**middle**) and associated potential drugs (**right**) for (**e**) the mRNAsi-related network and (**f**) the mDNAsi-related network.

**Figure 5 ijms-26-11684-f005:**
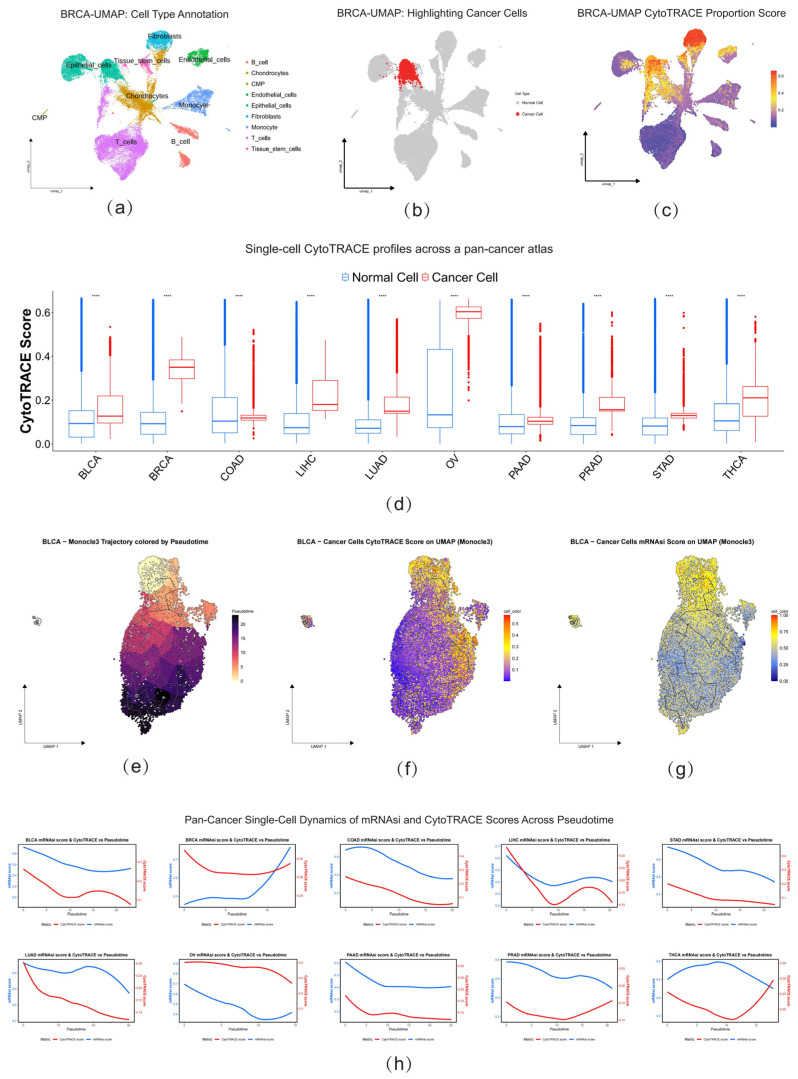
Single-Cell Sequencing Reveals Tumor Stemness Heterogeneity and Dynamic Trajectories. (**a**) UMAP plot showing cell-type annotation for all cells in the BRCA single-cell dataset. (**b**) UMAP plot highlighting the identified malignant cancer cell population in BRCA. (**c**) UMAP plot colored by the CytoTRACE stemness score in BRCA, with higher scores indicating greater stemness. (**d**) Pan-cancer comparison of CytoTRACE scores between normal cells (blue) and cancer cells (red) across 10 cancer types from the single-cell atlas. Statistical significance between the two groups was determined using the Wilcoxon test. Asterisks indicate the level of statistical significance (****: *p* < 0.0001). (**e**) Pseudotime trajectory analysis of malignant cells in BLCA using Monocle3, with cells colored by their position along the inferred differentiation path. (**f**) Visualization of CytoTRACE scores on the BLCA pseudotime trajectory UMAP. (**g**) Visualization of mRNAsi scores on the BLCA pseudotime trajectory UMAP. (**h**) Line plots showing the dynamic changes of CytoTRACE (red line) and mRNAsi (blue line) scores along the pseudotime trajectory across 10 different cancer types.

**Figure 6 ijms-26-11684-f006:**
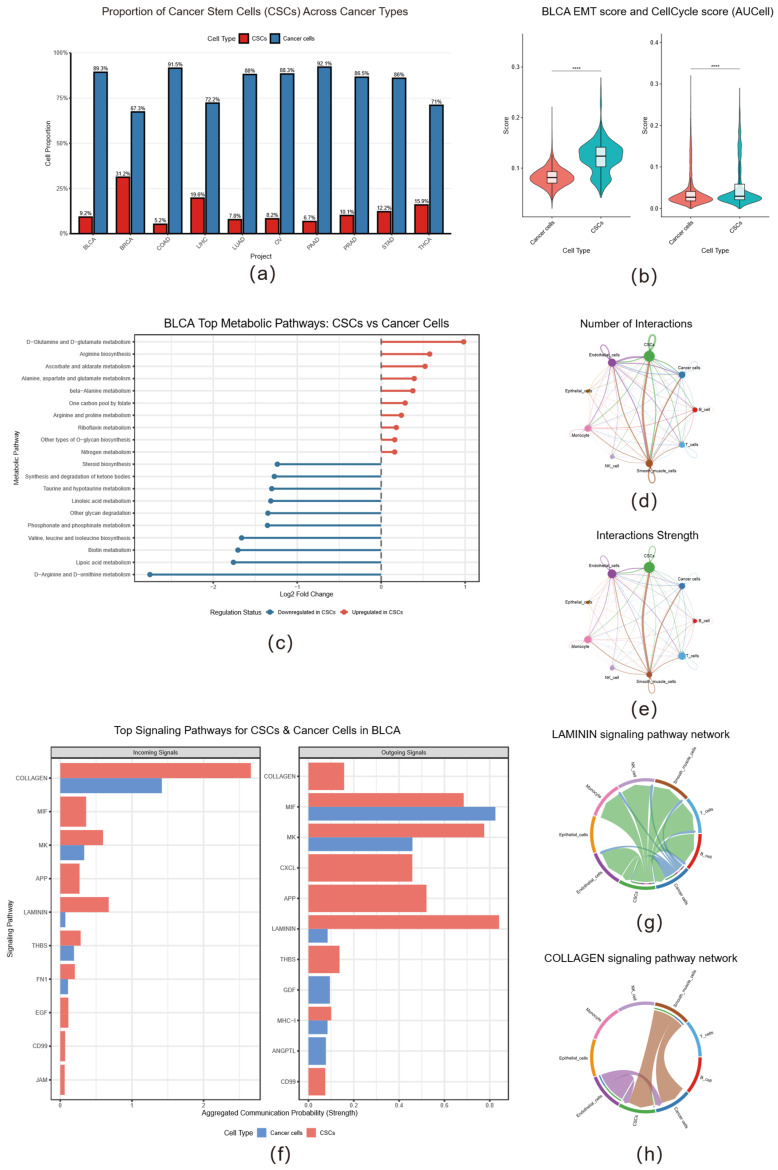
Analysis of functional characteristics, metabolic profiles, and communication patterns of cancer stem cells (CSCs). (**a**) Bar plot showing the proportion of CSCs (red) versus ordinary cancer cells (blue) across multiple cancer types. (**b**) AUCell analysis scores for epithelial–mesenchymal transition (EMT) and cell-cycle activity in BLCA. CSCs exhibit significantly higher EMT and cell-cycle activity compared to ordinary cancer cells (**** *p* < 0.0001, Wilcoxon rank-sum test). (**c**) Bar plot showing the most significantly different metabolic pathways in CSCs relative to ordinary cancer cells in bladder cancer (BLCA). Red indicates pathways upregulated in CSCs, while blue represents downregulated pathways. (**d**,**e**) Circle plots visualizing the cell–cell communication network in the tumor microenvironment, showing (**d**) the total number of interactions and (**e**) the total interaction strength, respectively. (**f**) Bar plot comparing the aggregated communication probability (strength) of major incoming (**left**) and outgoing (**right**) signaling pathways for CSCs versus ordinary cancer cells in BLCA. CSCs show stronger communication activity in key signaling pathways such as COLLAGEN, MIF, and LAMININ. (**g**,**h**) Visualization of key signaling pathway networks: (**g**) the LAMININ pathway with CSCs as the core signal sender, and (**h**) the COLLAGEN pathway with CSCs as the core signal receiver.

**Figure 7 ijms-26-11684-f007:**
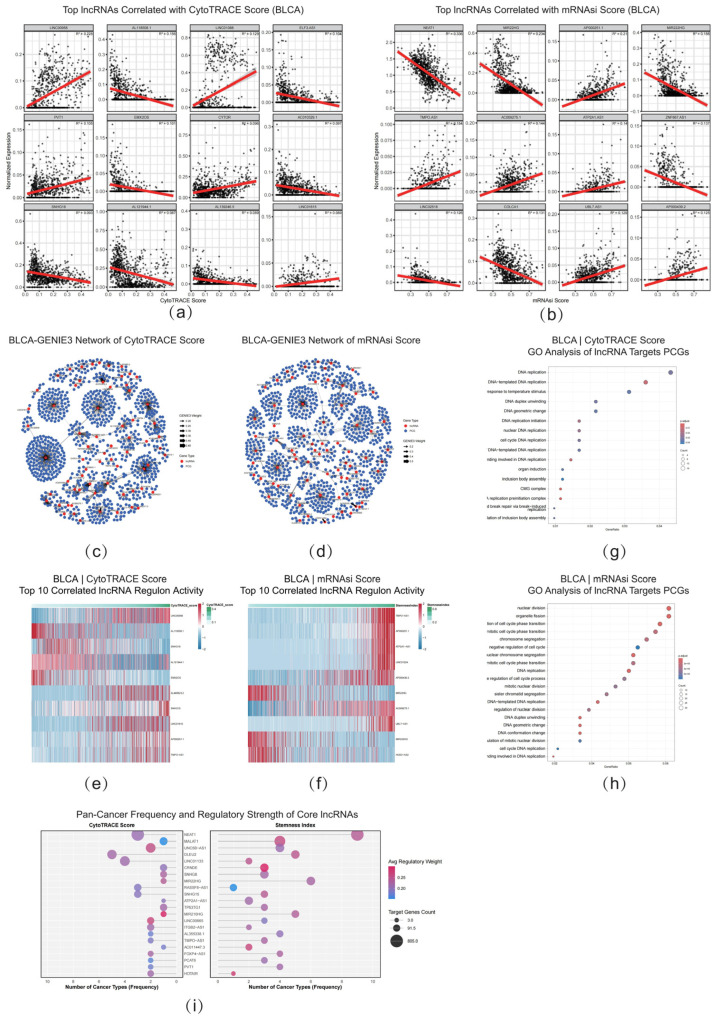
Single-Cell Transcriptome Reveals Core lncRNA Networks Regulating Tumor Stemness. (**a**) Scatter plots of top lncRNAs correlated with CytoTRACE Score in BLCA. (**b**) Scatter plots of top lncRNAs correlated with mRNAsi Score in BLCA. (**c**) GENIE3 regulatory network associated with CytoTRACE Score in BLCA. (**d**) GENIE3 regulatory network associated with mRNAsi Score in BLCA. (**e**) Heatmap of the top 10 correlated lncRNA regulon activities for CytoTRACE Score in BLCA. (**f**) Heatmap of the top 10 correlated lncRNA regulon activities for mRNAsi Score in BLCA. (**g**) GO enrichment analysis of lncRNA target PCGs related to CytoTRACE Score. (**h**) GO enrichment analysis of lncRNA target PCGs related to mRNAsi Score. (**i**) Pan-cancer frequency and regulatory strength of core lncRNAs, analyzed for CytoTRACE Score (**left**) and Stemness Index (**right**). Bubble size represents target gene count, and color indicates average regulatory weight.

## Data Availability

The datasets analyzed in this study are publicly available and can be found in the following public repositories. Bulk RNA sequencing data and corresponding clinical information were obtained from The Cancer Genome Atlas (TCGA) database (accessible at: https://portal.gdc.cancer.gov/, accessed on 5 September 2023). Single-cell RNA sequencing data were obtained from the Gene Expression Omnibus (GEO) database under the accession number GSE210347 (accessible at: https://www.ncbi.nlm.nih.gov/geo/, accessed on 26 December 2024).

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
