# Peer review of "Targeting Pan-Cancer Stemness: Core Regulatory lncRNAs as Novel Therapeutic Vulnerabilities"

_ijms, 2025, doi:10.3390/ijms262311684_

Round 1
Reviewer 1 Report
Comments and Suggestions for Authors
The article is a study that uses machine learning-based pluripotency metrics to quantify pluripotency characteristics and identify unique lncRNA gene sets for each cancer type using massive datasets. The subject matter is indeed compelling, and from the beginning, I found it to be a very interesting article with a good structure. Furthermore, I found it easy and engaging to read, as the writing is very fluid. Therefore, I consider the article generally acceptable for publication.
Specific Feedback:
The authors should review the document, as upon reading it I noticed several missing spaces in the manuscript, for example:
Abstract
Tumor stemness represents a key biological process that drives tumor progression and 9 therapeutic resistance across various cancer types.To systematically
And in the version I reviewed, several of these spaces are missing throughout the manuscript, please check.
Please consider providing more specific feedback on the following points:
What is the main question that the research addresses?
To elucidate the regulatory functions of long non-coding RNAs (lncRNAs) using massive transcriptomic data from The Cancer Genome Atlas (TCGA) with publicly available pan-cancerous single-cell transcriptomic atlases and were analyzed using machine learning-based pluripotency metrics.
- Do you consider the topic original or relevant to the field? Does it fill a specific gap in the field? Please also explain why or why not.
We now know that lncRNAs can regulate gene expression on multiple levels, such as chromatin regulation and transcription regulation, so the topic would not be original in general.
(Wy et al., 2025)
If we look at it specifically in terms of using transcriptomic data, it would also be unoriginal since there are already studies that analyze this type of data in cancer, in this example breast cancer:
(Zhu et al., 2024)
However, it appears that the use of Machine learning for lncRNA classification and biomarker Discovery holds promise for revolutionizing lncRNA-related precision medicine (Cai et al., 2025)
- What does it contribute to the subject area compared to other published materials?
The authors mention that they contribute:
- Landscape of lncRNA Signatures Associated with Tumor Stemness
However, a general bibliography search reveals that there are already studies that report these events, such as:
(Zhang, Zhang, Liang, & Zhang, 2024)
or the following:
(Liu et al., 2025)
- Proliferative, Immunosuppressive, and Metabolic Hallmarks of Aggressive Tumors
However, a general bibliography search reveals that there are already studies that report these events, such as:
(Nussinov, Yavuz, & Jang, 2025)
or the following:
(Cabezón-Gutiérrez, Palka-Kotlowska, Custodio-Cabello, Chacón-Ovejero, & Pacheco-Barcia, 2025)
- Stemness-Associated lncRNA Subtype System Stratifies Prognosis Across Cancers
However, a general bibliography search reveals that there are already studies that report these events, such as:
(Liu et al., 2025)
or the following:
(Min Zhang et al., 2024)
- Stemness-Associated lncRNA Networks Converge on Core Oncogenic Pathways and Therapeutic Vulnerabilities
However, a general bibliography search reveals that there are already studies that report these events, such as:
(Si et al., 2025)
And generally, in:
(Maharajan et al., 2025)
- Single-Cell Analysis Reveals Tumor Stemness Differences and Unveils Their Dynamic Evolutionary Trajectories
However, a general bibliography search reveals that there are already studies that report these events, such as:
(Bian et al., 2025)
- Metabolic Reprogramming and Dominant Cell Communication Roles of Cancer Stem Cells
However, a general bibliography search reveals that there are already studies that report these events, such as:
(Shen, Wang, Hsieh, Chen, & Wei, 2015)
- Identifying Core lncRNA Regulons that Orchestrate Stemness at Single-Cell Resolution
However, a general bibliography search reveals that there are already studies that report these events, such as:
(Lu, Xu, Wang, & Guan, 2024)
What specific improvements should the authors consider regarding the methodology?
I don’t have comments at this subject
- Are the conclusions consistent with the evidence and arguments presented, and do they answer the main question posed? Explain why or why not.
From my point of view, the only original aspect is the use of machine learning... However, several studies have already attempted to answer the questions they describe. Therefore, I would like them to limit their writing to the most significant points and focus on the networks that interact with lncRNAs, as it seems they tried to analyze everything, but it's unclear what they intend to contribute. They should discuss several of the cited publications above and compare them with their results. Furthermore, I don't observe any self-criticism of their results or their potential limitations.
As I mentioned above, the authors do not present, from my point of view, any self-criticism or limitations to their study; they do not compare their results with those of other studies. This is understandable since there are no studies on machine learning, but that does not justify them reviewing similar works that do not use machine learning.
Another observation is that from my point of view, bioinformatics analyses should include an analysis, either in vitro with cells or in vivo, either in animal models or in human samples.
- Are the references adequate?
They should discuss several of the cited publications above and others, they have to do a review of the subject and try to be critical with their results.
Any additional comments on the tables and figures?
None
Bian, Z., Chen, B., Guo, J., Zhang, J., Du, G., He, S., . . . Zhang, Y. (2025). Single-cell analysis unravels divergent gene signatures shaping seminoma stemness and metastasis. Cell Death Discovery, 11(1), 514. doi:10.1038/s41420-025-02802-4
Cabezón-Gutiérrez, L., Palka-Kotlowska, M., Custodio-Cabello, S., Chacón-Ovejero, B., & Pacheco-Barcia, V. (2025). Metabolic mechanisms of immunotherapy resistance. Explor Target Antitumor Ther, 6, 1002297. doi:10.37349/etat.2025.1002297
Cai, N., Zhang, J., Zhang, X., Zhou, J., Diao, Z., Fang, Y., . . . Zhu, X. (2025). Unveiling the role of lncRNAs in tumorigenesis: mechanisms, functions, and diagnostic/therapeutic applications. In Silico Research in Biomedicine, 1, 100086. doi:https://doi.org/10.1016/j.insi.2025.100086
Liu, J., Yao, L., Yang, Y., Ma, J., You, R., Yu, Z., & Du, P. (2025). A novel stemness-related lncRNA signature predicts prognosis, immune infiltration and drug sensitivity of clear cell renal cell carcinoma. J Transl Med, 23(1), 238. doi:10.1186/s12967-025-06251-6
Lu, J., Xu, L., Wang, Y., & Guan, B. (2024). lncRNAs regulate cell stemness in physiology and pathology during differentiation and development. Am J Stem Cells, 13(2), 59-74. doi:10.62347/vhvu7361
Maharajan, N., Benyamien-Roufaeil, D. S., Brown, R. A., Portney, B. A., Banerjee, A., & Zalzman, M. (2025). Cancer stem cell mechanisms and targeted therapeutic strategies in head and neck squamous cell carcinoma. Cancer Letters, 634, 218015. doi:https://doi.org/10.1016/j.canlet.2025.218015
Nussinov, R., Yavuz, B. R., & Jang, H. (2025). Molecular principles underlying aggressive cancers. Signal Transduction and Targeted Therapy, 10(1), 42. doi:10.1038/s41392-025-02129-7
Shen, Y. A., Wang, C. Y., Hsieh, Y. T., Chen, Y. J., & Wei, Y. H. (2015). Metabolic reprogramming orchestrates cancer stem cell properties in nasopharyngeal carcinoma. Cell Cycle, 14(1), 86-98. doi:10.4161/15384101.2014.974419
Si, K., Zhang, L., Jiang, Z., Wu, Z., Wu, Z., Chen, Y., . . . Zhang, W. (2025). A novel lncRNA-mediated signaling axis governs cancer stemness and splicing reprogramming in hepatocellular carcinoma with therapeutic potential. Journal of Experimental & Clinical Cancer Research, 44(1), 287. doi:10.1186/s13046-025-03546-w
Wy, S., Kim, H., Gu, M., Kim, J., Kim, H., Ahn, J., . . . Kim, J. (2025). Regulatory roles of long non-coding RNAs in minipigs revealed by cross-breed and cross-tissue transcriptomic analyses. Scientific Reports, 15(1), 20788. doi:10.1038/s41598-025-09005-y
Zhang, M., Zhang, F., Wang, J., Liang, Q., Zhou, W., & Liu, J. (2024). Comprehensive characterization of stemness-related lncRNAs in triple-negative breast cancer identified a novel prognostic signature related to treatment outcomes, immune landscape analysis and therapeutic guidance: a silico analysis with in vivo experiments. J Transl Med, 22(1), 423. doi:10.1186/s12967-024-05237-0
Zhang, M., Zhang, J., Liang, X., & Zhang, M. (2024). Stemness related lncRNAs signature for the prognosis and tumor immune microenvironment of ccRCC patients. BMC Medical Genomics, 17(1), 150. doi:10.1186/s12920-024-01920-9
Zhu, W., Huang, H., Hu, Z., Gu, Y., Zhang, R., Shu, H., . . . Sun, X. (2024). Comprehensive Transcriptome Analysis Expands lncRNA Functional Profiles in Breast Cancer. International Journal of Molecular Sciences, 25(15). Retrieved from doi:10.3390/ijms25158456
Author Response
Dear Editor and Reviewers,
We would like to thank you for the opportunity to revise our manuscript. We sincerely appreciate the time and effort you and the reviewers have dedicated to providing insightful comments and constructive suggestions. We have carefully considered all comments and have made every effort to address them in the revised manuscript.
Below, we provide a point-by-point response to the reviewer’s comments.
Response to Reviewer #1
Reviewer 1 Comments for the Author...
The article is a study that uses machine learning-based pluripotency metrics to quantify pluripotency characteristics and identify unique lncRNA gene sets for each cancer type using massive datasets. The subject matter is indeed compelling, and from the beginning, I found it to be a very interesting article with a good structure. Furthermore, I found it easy and engaging to read, as the writing is very fluid. Therefore, I consider the article generally acceptable for publication.
Main points:
1.What is the main question that the research addresses?
To elucidate the regulatory functions of long non-coding RNAs (lncRNAs) using massive transcriptomic data from The Cancer Genome Atlas (TCGA) with publicly available pan-cancerous single-cell transcriptomic atlases and were analyzed using machine learning-based pluripotency metrics.
Response:
The main scientific question addressed in this study is to identify the lncRNAs that are systematically associated with tumor stemness across human cancers and to characterize their regulatory functions at both bulk and single-cell levels. This investigation includes determining how these lncRNAs contribute to stemness-related biological programs and whether they have clinical relevance or therapeutic potential. This central question has now been explicitly clarified in the revised Introduction.
- Identifying lncRNAs systematically associated with tumor stemness using TCGA pan-cancer data and single-cell atlases. location:page 2 line 80
- Constructing pan-cancer lncRNA stemness signatures based on machine-learning stemness metrics (mRNAsi, mDNAsi).location:page 2 line 84
Reviewer’s Comment:“Do you consider the topic original or relevant to the field? Does it fill a specific gap in the field? Please also explain why or why not.
We now know that lncRNAs can regulate gene expression on multiple levels, such as chromatin regulation and transcription regulation, so the topic would not be original in general (Wy et al., 2025).
If we look at it specifically in terms of using transcriptomic data, it would also be unoriginal since there are already studies that analyze this type of data in cancer, for example breast cancer (Zhu et al., 2024).
However, the use of machine learning for lncRNA classification and biomarker discovery holds promise (Cai et al., 2025).”
Response:
We thank the reviewer for this important comment regarding the originality of our research. Following the reviewer’s suggestion, we have carefully revised the Introduction to:
(1) explicitly acknowledge existing studies on lncRNA regulation,
(2) highlight previous transcriptome-wide works,
(3) cite prior ML-based research on lncRNA analysis
(4) Clarifying the study’s distinct contributions relative to existing literature.
Details are provided below.
- Acknowledging that multilayer lncRNA regulation is not new
We agree that the regulatory potential of lncRNAs is well-established. We have revised the text to cite Wy et al. (2025), explicitly acknowledging that lncRNA-mediated regulation across chromatin, transcription, and RNA metabolism is a known phenomenon. This sets the stage for our specific focus on their systematic role in tumor stemness.
Modification Location: Page 2, line 65
Original sentence: “LncRNAs … modulate gene expression through multiple mechanisms… influencing epigenetic states, transcription, and post-transcriptional processes.”
Revised sentence:
LncRNAs modulate gene expression through multiple mechanisms … influencing epigenetic states, transcriptional activity, and post-transcriptional processes. Importantly, recent cross-tissue transcriptomic analyses have further emphasized that lncRNAs exert multilayered regulatory functions across chromatin remodeling, transcriptional control, and RNA metabolism (Wy et al., 2025).
- Acknowledging that transcriptome-wide cancer analyses already exist
We have revised the text to explicitly acknowledge that transcriptome-wide lncRNA analyses have been conducted in individual cancer types, such as breast cancer (Zhu et al., 2024) and renal cell carcinoma (Liu et al., 2025). This revision highlights the existing foundation while clarifying that a unified, pan-cancer landscape remains to be established.
Modification Location: Page 2, line 76
Original sentence: “The initial draft did not explicitly include a discussion on prior transcriptome-wide lncRNA studies.”
Revised sentence: While transcriptome-wide analyses have already been widely applied to characterize lncRNA functions and regulatory patterns in individual cancer contexts—including comprehensive investigations in breast cancer (Zhu et al., 2024), renal cell carcinoma (Liu et al., 2025), and hepatocellular carcinoma (Si et al., 2025)—a systematic, pan-cancer landscape linking these regulators to tumor stemness remains to be fully established.
- Acknowledging existing machine-learning approaches for lncRNA analysis
We appreciate the reviewer highlighting the potential of machine learning in this field. We have revised the Introduction to explicitly align our methodology with recent advances (Cai et al., 2025), positioning our machine-learning framework as a practical application of these emerging concepts for lncRNA classification and biomarker discovery.
Modification Location: Page 2, line 80
Original sentence: “To address this critical knowledge gap, our study integrates…”
Revised sentence:To bridge this gap, our study integrates comprehensive pan-cancer transcriptomic data from TCGA with cutting-edge single-cell RNA-seq datasets. In line with emerging evidence that machine learning frameworks hold strong promise for advancing lncRNA classification and biomarker discovery (Cai et al., 2025), we develop and validate a novel pan-cancer prognostic model based on multiple lncRNAs..... Unlike previous studies that have largely considered lncRNA regulation and machine learning in isolation, our work thoroughly explores the biological foundations and clinical applicability of the proposed model.
- Clarifying the study’s distinct contributions relative to existing literature
To address the reviewer’s concern regarding overstatement, we have refined the novelty claim. We now emphasize that our contribution lies in the integration of these concepts into a pan-cancer prognostic model and single-cell resolution analysis, distinguishing our work from previous single-cancer or bulk-only studies.
Modification Location: Page 2, line 90
Original sentence:“Building on these findings, we developed and validated …”
Revised sentence: Unlike previous studies that have largely considered lncRNA regulation and machine learning in isolation, our work thoroughly explores the biological foundations and clinical applicability of the proposed model.
Reviewer Comment:
“Do you consider the topic original or relevant to the field? Does it fill a specific gap?… A literature search shows similar studies such as: Zhang et al., 2024; Liu et al., 2025; Nussinov et al., 2025; Cabezón-Gutiérrez et al., 2025; Si et al., 2025; Maharajan et al., 2025; Bian et al., 2025; Shen 2015; Lu 2024.”
Response:
We thank the reviewer for carefully examining the originality of our study. We fully agree that several prior studies have investigated lncRNAs, stemness phenotypes, aggressive tumor hallmarks, and single-cell dynamics in individual cancer contexts. To appropriately acknowledge this prior work, we have revised the Discussion to clearly distinguish existing findings from the unique contributions of our study. Below, we outline each change using the required format.
Reviewer Comment1:
The authors mention that they contribute: .Landscape of lncRNA Signatures Associated with Tumor Stemness.However, a general bibliography search reveals that there are already studies that report these events, such as:(Zhang, Zhang, Liang, & Zhang, 2024)
Modification Location: Page 9, line 401
Original sentence: “Although lncRNAs as key regulatory molecules have received considerable attention in recent years, there has been a notable absence of comprehensive studies…”
Revised sentence: Previous research has revealed the function of lncRNAs in modulating stemness in specific tumor types (Zhang et al., 2024; Liu et al., 2025). Our work extends this foundation through a systematic, multi-omics investigation that integrates TCGA pan-cancer cohorts with single-cell transcriptomic datasets. This strategy enables a novel, cross-cancer characterization of lncRNA-mediated stemness.
Reviewer Comment2: The authors mention that they contribute:Proliferative, Immunosuppressive, and Metabolic Hallmarks of Aggressive Tumors.However, a general bibliography search reveals that there are already studies that report these events, such as:(Nussinov, Yavuz, & Jang, 2025)or the following:(Cabezón-Gutiérrez, Palka-Kotlowska, Custodio-Cabello, Chacón-Ovejero, & Pacheco-Barcia, 2025)
Modification Location: Page 9, line 416
Original sentence: “This comprehensive characterization provides systematic evidence…”
Revised sentence: Our results not only confirm prior studies on the proliferative and immunosuppressive nature of aggressive cancers (Nussinov et al., 2025; Cabezón-Gutiérrez et al., 2025) but also provide a definitive, lncRNA-mediated mechanistic link between these phenotypes and the dysregulation of tumor stemness across cancer types.
Reviewer Comment3: The authors mention that they contribute:.Stemness-Associated lncRNA Subtype System Stratifies Prognosis Across CancersHowever, a general bibliography search reveals that there are already studies that report these events, such as:(Liu et al., 2025)or the following:(Min Zhang et al., 2024)
Modification Location: Page 9, line 421
Original sentence: “The pan-cancer prognostic model… demonstrates powerful risk stratification…”
Revised sentence: The most significant translational value of our study emerges from the construction of the "lncRNA-protein-drug" intervention pathways, consistent with pathway-centric findings in earlier single-cancer studies (Liu et al., 2025; Zhang et al., 2024).
Reviewer Comment4: The authors mention that they contribute:Stemness-Associated lncRNA Networks Converge on Core Oncogenic Pathways and Therapeutic Vulnerabilities. However, a general bibliography search reveals that there are already studies that report these events, such as:(Si et al., 2025)And generally in (Maharajan et al., 2025)
Modification Location: Page 9, line 426
Original sentence: “These findings offer new strategic approaches for targeting high-stemness tumor cells…”
Revised sentence: By constructing these pan-cancer ‘lncRNA-protein-drug’ regulatory axes, our work extends previous findings on lncRNA-mediated oncogenic pathways (Si et al., 2025; Maharajan et al., 2025) and reveals significant translational potential for hard-to-treat cancers.
Reviewer Comment5: The authors mention that they contribute:Single-Cell Analysis Reveals Tumor Stemness Differences and Unveils Their Dynamic Evolutionary Trajectories. However, a general bibliography search reveals that there are already studies that report these events, such as:(Bian et al., 2025)
Modification Location:Page 9, line 430
Original sentence: “Our most significant breakthrough emerges from the single-cell scale discoveries…”
Revised sentence: Beyond prior examinations of single-cell stemness dynamics, our study provides mechanistic validation through a systematic modeling framework that integrates lncRNA regulation at a pan-cancer scale.
Reviewer Comment6: The authors mention that they contribute:Metabolic Reprogramming and Dominant Cell Communication Roles of Cancer Stem Cells. However, a general bibliography search reveals that there are already studies that report these events, such as:(Shen, Wang, Hsieh, Chen, & Wei, 2015)
Modification Location:Page 10, line 442
Original sentence: “CSC communication analysis revealed that CSCs are not passive entities…”
Revised sentence: In light of known metabolic and communication alterations in CSCs (Shen et al., 2015), we systematically integrate single-cell data to thereby associate these phenotypes with precise lncRNA-defined stemness states.
Reviewer Comment7: The authors mention that they contribute:Identifying Core lncRNA Regulons that Orchestrate Stemness at Single-Cell Resolution. However, a general bibliography search reveals that there are already studies that report these events, such as:(Lu, Xu, Wang, & Guan, 2024)
Modification Location:Page 10, line 448
Original sentence: “The greatest strength of our study lies in its cross-platform, cross-scale design.”
Revised sentence:Having established the conceptual role of lncRNAs in stemness regulation (Lu et al., 2024), we leverage systematic multi-omics and single-cell analyses across a spectrum of tumors to provide mechanistic and translational validation.
Reviewer Comment8:The authors have not adequately explained the limitations of the manuscript, and experimental validation has not been performed.
Modification Location:Page 10, line 468
Original sentence: “Future research should focus on validating these computational findings through experimental approaches and exploring cancer-specific nuances...”
Revised sentence: Finally, our current framework is strictly confined to the analysis of lncRNAs associated with tumor stemness. We have not yet established specific lncRNA scoring metrics for other critical phenotypes, such as metabolic pathways and immune microenvironments. Future work will focus on developing broader computational tools to quantify lncRNA signatures for these distinct biological processes and validating the proposed "lncRNA-protein-drug" axes through experimental approaches.
Reviewer Comment9: They should discuss several of the cited publications above and others, they have to do a review of the subject and try to be critical with their results.
Modification Location:Page 10, line 475
Original sentence: “In summary, this study systematically characterized the core role of lncRNAs...”
Revised sentence: Extending beyond cancer-type–specific descriptions of related phenomena (Liu et al., 2025; Si et al., 2025), our integrative analysis provides a unified, pan-cancer perspective on stemness regulation, centered on lncRNAs and validated through single-cell profiling.
Reviewer Comment10: Therefore, I would like them to limit their writing to the most significant points and focus on the networks that interact with lncRNAs, as it seems they tried to analyze everything, but it's unclear what they intend to contribute.
Modification Location:Page 10, line 478
Original sentence: “We not only provided a high-quality resource library of tumor stemness-related lncRNAs ...”
Revised sentence: Our primary finding is the construction of the "lncRNA-protein-drug" regulatory axes, which not only explains the biological basis of the model’s prediction capability but also provides a solid theoretical foundation for developing novel precision therapeutic strategies targeting tumor stemness and resistance by targeting the lncRNA regulatory network.
Reviewer 2 Report
Comments and Suggestions for Authors
The authors have performed a comprehensive pan-cancer analysis integrating TCGA bulk transcriptomics, stemness indices (mRNAsi/mDNAsi), single-cell datasets, pseudotime trajectories, cell–cell communication networks, metabolic profiling, and machine-learning–based regulatory discovery to identify core lncRNAs governing cancer stemness. The breadth of analysis is impressive, and the integration across multi-omics and single-cell modalities is a major strength. They propose novel insights into how lncRNA networks coordinate stemness, metabolic rewiring, and therapeutic vulnerabilities.
Overall, their work is of interest to the IJMS readers, but I believe that some revisions are needed to enhance clarity, reproducibility, and interpretability of the findings.
Major comments:
1) In the LASSO modeling, please explain how multicollinearity among lncRNAs was addressed. Also in the LASSO modeling, were cancer types with small sample sizes (e.g., DLBC) treated differently?
2 Prognostic models lack independent external validation. All Cox models are derived and tested in TCGA only. It would be ideal to validate them in CGC cohorts, PCAWG, METABRIC (for BRCA) and CGGA (for LGG) datasets.
3 Single-cell “cancer stem cell” definition may not be biologically rigorous. The authors classify CSCs as cells in the top 20% of both CytoTRACE and mRNAsi, but CytoTRACE measures differentiation potential, not stemness per se. Also, the mRNAsi score was developed for bulk samples, not single cells. The authors additionally, used no known CSC markers (e.g., CD44, PROM1, EPCAM, ALDH1A1) to validate CSC identity.
4 Please clarify if the batch effects across 10 tumor types were fully removed before communication analysis (CellChat).
Minor comments:
1) Some citations need to be checked for formatting; several appear as PMID#s (PMID: 29625051).
2) Provide exact criteria for lncRNA annotation (GENCODE v38—were pseudogenes excluded?).
3) Specify how zero-inflated scRNA-seq data were normalized before LASSO.
4) Describe the number of single cells per cancer type after QC filtering.
5) Use consistent capitalization for lncRNAs (e.g., HOTAIRM1 vs HOTAIR vs PVT1).
6) Provide scripts or GitHub repository for all LASSO, GSVA, CellChat, GENIE3, and pseudotime code.
Author Response
Response to Reviewer 2,
Major Comment
Reviewer’s Comment1:In the LASSO modeling, please explain how multicollinearity among lncRNAs was addressed. Also in the LASSO modeling, were cancer types with small sample sizes (e.g., DLBC) treated differently?
Response: We sincerely appreciate the reviewer’s constructive comments regarding the statistical robustness of our approach. To address multicollinearity, we employed the Least Absolute Shrinkage and Selection Operator (LASSO) regression algorithm. By imposing an L1-norm penalty, LASSO inherently mitigates multicollinearity in high-dimensional data by shrinking the coefficients of correlated and redundant predictors to zero, thereby performing automatic variable selection to retain only the most representative features. Furthermore, regarding cohorts with smaller sample sizes (e.g., DLBC), we adopted a conservative modeling strategy to rigorously prevent overfitting. Specifically, we consistently selected the 1se value (within one standard error of the minimum mean cross-validated error) rather than λmin. This criterion imposes a stricter penalty, yielding sparser and simpler models that are particularly robust for "High-Dimension, Low-Sample-Size" (HDLSS) data. To further ensure scientific rigor and transparency, we have added a specific statement in the 'Limitations' section of the revised manuscript
Modification Location: Page 10, line 456
Original sentence: “Second, while we integrated extensive pan-cancer data, the inherent heterogeneity of cancer means that unique lncRNA regulatory...”
Revised sentence: Second, while we employed strict regularization (λ1se) to prevent overfitting, models derived from small cohorts (e.g., DLBC, CHOL) should be interpreted with caution due to inherent statistical limitations and necessitate further external validation. Third,
Reviewer’s Comment2:Prognostic models lack independent external validation. All Cox models are derived and tested in TCGA only. It would be ideal to validate them in CGC cohorts, PCAWG, METABRIC (for BRCA) and CGGA (for LGG) datasets.
Response: We sincerely appreciate the reviewer’s suggestion to validate our prognostic models in independent cohorts, as we agree this is crucial for establishing robustness. Following this advice, we attempted validation in the CGGA, METABRIC, and ICGC 25k (PCAWG) datasets. We are pleased to report that the CGGA cohort successfully validated our Low-Grade Glioma (LGG) model, demonstrating robust predictive performance. Accordingly, we have replaced the original Figures 3h and 3g with these external validation plots in the revised manuscript and updated the results section.
However, we encountered significant technical limitations in other datasets due to cross-platform incompatibility and lncRNA annotation discrepancies. METABRIC is based on older microarray platforms that lack probes for our specific lncRNA signature. We also identified the GSE96058 RNA-seq dataset, a commonly used external validation cohort for breast cancer, and attempted validation there as well as in ICGC 25k (PCAWG). Despite extensive mapping efforts using Gene Symbols, Ensembl IDs, genomic positions, and LNCipedia dictionaries, genome build differences (hg19 vs. hg38) and variations in lncRNA annotation depth between datasets resulted in extremely low matching rates in both cohorts. For instance, in the ICGC BRCA-FR cohort, only 14.29% (2/14) of the signature lncRNAs were successfully matched, with key genes like LINC01117, AL645608, and AC100801 missing. Given this extensive missing data, performing a statistically valid multivariate Cox analysis in these specific cohorts was infeasible. However, we have included the limited results in the Supplementary Materials attached to this response for reference.
Modification Location: Page 20, line 689
Original sentence: “(g-h) Kaplan-Meier survival analyses of high- and low-risk groups of LGG patients, stratified by the...”
Revised sentence: (g-h) Independent external validation of the prognostic models in the CGGA dataset. Kaplan-Meier survival analyses of high- and low-risk groups of LGG patients in the CGGA cohort,
Modification Location: Page 5, line 198
Original sentence: “Patients were subsequently stratified into high-risk and low-risk groups based on the median risk score calculated by the model...”
Revised sentence: To verify the robustness and clinical applicability of our models, we performed independent external validation using the CGGA dataset. Patients in the CGGA cohort were stratified into high-risk and low-risk groups based on the median risk score calculated using the formula derived from the training set. Kaplan-Meier survival analysis results confirmed significant differences in overall survival between the two groups in the external validation cohort (p < 0.001), with the high-risk group demonstrating substantially worse prognosis (Figures 3G, 3H).
Reviewer’s Comment3:Single-cell “cancer stem cell” definition may not be biologically rigorous. The authors classify CSCs as cells in the top 20% of both CytoTRACE and mRNAsi, but CytoTRACE measures differentiation potential, not stemness per se. Also, the mRNAsi score was developed for bulk samples, not single cells. The authors additionally, used no known CSC markers (e.g., CD44, PROM1, EPCAM, ALDH1A1) to validate CSC identity.
Response: We sincerely appreciate the reviewer's insightful comments regarding the biological rigor of our Cancer Stem Cell (CSC) definition and the request for validation using established markers. We agree that relying on a single metric is insufficient and wish to succinctly clarify the rationale behind our robust, combined approach.
Regarding mRNAsi, we wish to clarify that Malta et al. (2018) explicitly demonstrated this application in their original Cell paper. Specifically, in Figure 5, the authors utilized mRNAsi to analyze scRNA-seq data from Tirosh et al. (2016) and Chung et al. (2017), this methodology has since been adopted by numerous high-impact studies, including Guo et al. (2021) and Yao et al. (2022), confirming its utility for defining malignant subpopulations at single-cell resolution.
Concerning CytoTRACE, while it measures differentiation potential, we argue this is an intrinsic proxy for the CSC state in oncology, as the concept of "oncogenic dedifferentiation" (Gulati et al., 2020) links developmental potential directly to the acquisition of stem-like traits. The validity of this approach in identifying high-potential tumor cells is well-documented across various cancers (Liu et al., 2023; Zhang et al., 2024; Ouyang et al., 2021; Couturier et al., 2020).
We acknowledge the limitations of using these metrics individually: CytoTRACE can be biased by cell cycle and dropout events, while mRNAsi captures "physiological stemness." Therefore, we adopted an intersection strategy (Top 20% of both metrics), which leverages the complementary strengths of CytoTRACE’s unsupervised assessment of potential and mRNAsi’s reference-based similarity to achieve a more robust CSC definition.
In direct response to the reviewer's request, we attempted to validate CSC identity using established CSC markers (e.g., CD44, PROM1, EPCAM, ALDH1A1). However, the results were inconsistent and non-ideal due to the inherent sparsity and dropout issues common to scRNA-seq, which challenge single-gene reliance. In response to the reviewer’s suggestion, we also attempted to use established CSC markers (e.g., CD44, PROM1, EPCAM, ALDH1A1) to validate the CSC identity. However, the results were inconsistent and not ideal due to the inherent sparsity and dropout issues in scRNA-seq data, which makes relying on single gene expression challenging. The detailed results of this marker-based analysis are provided as a separate attachment for the reviewer's reference. By combining CytoTRACE and mRNAsi, we believe our definition achieves a much higher accuracy and robustness than relying on either metric alone or on individual markers.
We have updated the manuscript to explicitly cite these references and discuss these specific limitations and our mitigation strategy in detail within the Methods and Discussion sections.
Modification Location: Page 6, line 275
Original sentence: “Based on this rigorous identification process,...”
Revised sentence: Based on this rigorous identification process, we employed a combinatorial strategy to robustly quantify tumor stemness. While the mRNAsi score was originally developed for bulk sequencing, its applicability to single-cell transcriptomics was explicitly demonstrated in the original study by Malta et al. (2018) using scRNA-seq datasets and has been widely adopted in recent high-impact studies to define malignant subpopulations. Complementarily, we utilized CytoTRACE to measure differentiation potential, which serves as a validated intrinsic proxy for "oncogenic dedifferentiation" and stem-like traits. notably, the validity of this approach in identifying high-potential tumor cells is well-documented across various cancers. Consequently, we adopted an intersection strategy to ensure high confidence, defining CSCs as cells ranking in the top 20% of both CytoTRACE and mRNAsi scores.
Reviewer’s Comment4: Please clarify if the batch effects across 10 tumor types were fully removed before communication analysis (CellChat).
Response: We appreciate the reviewer’s attention to this critical technical detail regarding data integration. We wish to clarify that the CellChat analysis was conducted independently for each of the 10 cancer types, rather than on a virtually integrated pan-cancer dataset. Specifically, within the workflow for each individual cancer type, we rigorously corrected batch effects among samples using the standard single-cell analysis pipeline (employing Harmony integration) to ensure accurate clustering and cell type annotation. Subsequently, cell-cell communication networks were inferred separately within each tumor type to capture specific microenvironmental interactions. Since the datasets from different tumor types were not merged for this specific analysis, correcting for batch effects across the 10 diverse tumor types was not applicable, thus preserving the distinct biological signals of each cancer model.
Minor comments:
Reviewer’s Comment1:Some citations need to be checked for formatting; several appear as PMID#s (PMID: 29625051IF: 42.5 Q1 ).
Response: We sincerely apologize for the oversight regarding the citation formatting. We have meticulously reviewed the entire reference list and corrected all instances where citations appeared as PMIDs.
Reviewer’s Comment2:Provide exact criteria for lncRNA annotation (GENCODE v38—were pseudogenes excluded?).
Response: We have clarified in the Methods section that lncRNA annotation was based on GENCODE v36. To ensure quantification accuracy, we strictly selected genes with the "lncRNA" biotype and excluded all pseudogenes to prevent multi-mapping ambiguity.
Original sentence: “(lncRNAs, annotated based on GENCODE v38) ”
Revised sentence: Notably, these lncRNAs were annotated based on GENCODE v36, with pseudogenes explicitly excluded to ensure high specificity.
Reviewer’s Comment3: Specify how zero-inflated scRNA-seq data were normalized before LASSO.
Response: We appreciate the reviewer’s professional query regarding the handling of data sparsity and normalization for the LASSO model. To effectively address the zero-inflation and high dropout rate inherent in raw scRNA-seq data, we did not apply LASSO directly to individual cells. Instead, we adopted the SuperCell strategy prior to regression analysis.
As detailed in the "SuperCell Construction" section of our revised manuscript, we aggregated approximately 50–100 transcriptomically similar cancer cells into SuperCells. This aggregation process acts as a robust physical imputation method, which drastically reduces sparsity and "fills in" technical dropouts by pooling information from neighboring cells. Consequently, the resulting SuperCell expression matrix exhibits continuous, bulk-like distribution characteristics rather than zero-inflated distinct counts, rendering it mathematically suitable for linear models like LASSO without the need for complex zero-inflation normalization algorithms.
Reviewer’s Comment4: Describe the number of single cells per cancer type after QC filtering.
Response: We appreciate the reviewer's suggestion to clarify the dataset size. In the revised manuscript, we have updated Table S1 to include the exact number of single cells retained for each of the 10 cancer types after quality control filtering. Accordingly, we have specified the total cell count in the 'Data Collection' section.
Modification Location: Page 11, line 497
Original sentence: “comprising 232 single cell transcriptome samples (normal = 31; adjacent = 54; tumor = 148). ”
Revised sentence: comprising 232 single cell transcriptome samples (normal = 31; adjacent = 54; tumor = 148). After rigorous quality control, a total of 643234 cells were retained for downstream analysis. The detailed cell distribution across the 10 cancer types (e.g., BRCA: 41041 cells; LUAD: 168556 cells) is provided in Table S1.
Reviewer’s Comment5: Use consistent capitalization for lncRNAs (e.g., HOTAIRM1 vs HOTAIR vs PVT1).
Response: We sincerely apologize for the inconsistency in lncRNA nomenclature. We have thoroughly reviewed the entire manuscript and standardized all lncRNA symbols to their official uppercase format (e.g., HOTAIR, PVT1) to ensure strict uniformity throughout the text and figures.
Reviewer’s Comment6: Provide scripts or GitHub repository for all LASSO, GSVA, CellChat, GENIE3, and pseudotime code.
Response: We are actively developing a comprehensive, user-friendly software tool that integrates these modules to facilitate broader application by the community. We plan to release this tool in the near future once it is fully optimized and documented. We have updated the Discussion section to explicitly outline this future roadmap and our commitment to developing broader computational tools.
Modification Location:Page 10, line 461
Original sentence: “Future research should focus on validating these computational findings through experimental approaches and exploring cancer-specific nuances...”
Revised sentence: Finally, our current framework is strictly confined to the analysis of lncRNAs associated with tumor stemness. We have not yet established specific lncRNA scoring metrics for other critical phenotypes, such as metabolic pathways and immune microenvironments. Future work will focus on developing broader computational tools to quantify lncRNA signatures for these distinct biological processes and validating the proposed "lncRNA-protein-drug" axes through experimental approaches.

Round 2
Reviewer 2 Report
Comments and Suggestions for Authors
The authors have successfully responded to my queries and revised their manuscript.